# Formation of organic color centers in air-suspended carbon nanotubes using vapor-phase reaction

Daichi Kozawa [1✉], Xiaojian Wu [2], Akihiro Ishii[1,3], Jacob Fortner[2], Keigo Otsuka [3], Rong Xiang [4,5], Taiki Inoue [5,6], Shigeo Maruyama [5], YuHuang Wang [2,7] & Yuichiro K. Kato [1,3✉]

Organic color centers in single-walled carbon nanotubes have demonstrated exceptional ability to generate single photons at room temperature in the telecom range. Combining the color centers with pristine air-suspended nanotubes would be desirable for improved performance, but all current synthetic methods occur in solution which makes them incompatible. Here we demonstrate the formation of color centers in air-suspended nanotubes using a vapor-phase reaction. Functionalization is directly verified by photoluminescence spectroscopy, with unambiguous statistics from more than a few thousand individual nanotubes. The color centers show strong diameter-dependent emission, which can be explained with a model for chemical reactivity considering strain along the tube curvature. We also estimate the defect density by comparing the experiments with simulations based on a one-dimensional exciton diffusion equation. Our results highlight the influence of the nanotube structure on vapor-phase reactivity and emission properties, providing guidelines for the development of high-performance near-infrared quantum light sources.

[1] Quantum Optoelectronics Research Team, RIKEN Center for Advanced Photonics, Saitama 351-0198, Japan. [2] Department of Chemistry and Biochemistry, University of Maryland, College Park, MD 20742, USA. [3] Nanoscale Quantum Photonics Laboratory, RIKEN Cluster for Pioneering Research, Saitama 351-0198, Japan. [4] State Key Laboratory of Fluid Power and Mechatronic Systems, School of Mechanical Engineering, Zhejiang University, Hangzhou 310027, China. [5] Department of Mechanical Engineering, The University of Tokyo, Tokyo 113-8656, Japan. [6] Department of Applied Physics, Osaka University, Osaka 565-0871, Japan. [7] Maryland NanoCenter, University of Maryland, College Park, MD 20742, USA. ✉email: daichi.kozawa@riken.jp; yuichiro.kato@riken.jp

Quantum technologies offer various advantages beyond the classical limits in secure communications[1], parallel computing[2], and sensing[3]. Solid-state single-photon sources[4] are a fundamental component in these technologies, and considerable progress has been made in various systems including quantum dots[5], diamond[6], SiC[7], and two-dimensional materials[8]. Of practical interest are single-walled carbon nanotubes (SWCNTs), since operation at room temperature and in the telecom range is possible. In particular, organic color centers formed on nanotubes[9] offer additional advantages with their optical properties being chemically tunable using a variety of molecular precursors, including aryl-halides[10–13], diazonium-salt[14–22], ozone[23–25], and hypochlorite[26] that can covalently bond to the carbon lattice. By introducing dopant states with different emission energies and achieving potential traps deeper than the thermal energy, single-photon sources with desired properties can be produced[27].

Further development of quantum emitters with improved performance is expected if color centers can be introduced to as-grown air-suspended SWCNTs known for their pristine nature[28,29]. In comparison to solution-processed tubes that have naturally formed quenching sites[30], the air-suspended nanotubes can be considered defect free except for the tube ends[28,29]. Such a system should provide an ideal platform for investigating photophysics of color centers[14,24], opening up new opportunities in nanoscale photonics using one- and zero-dimensional hybrid structures. Existing methods, however, require liquid-phase reactions where solvents and surfactants are inevitably in contact with the nanotubes, making the reactions incompatible with air-suspended nanotubes. To combine the excellent optical properties of the air-suspended SWCNTs with these organic color centers, an intelligent design of chemical reactions is needed.

In this work, we propose and demonstrate a vapor-phase reaction to create organic color centers in air-suspended SWCNTs. Tubes are functionalized with a photochemical reaction where adsorbing precursor vapor allows for preserving the suspended structures because of a weak mechanical perturbation. Individual tubes are characterized by confocal microspectroscopy to verify the formation of color centers, and we conduct a statistical survey of more than 2000 photoluminescence (PL) spectra to investigate diameter-dependent emission intensities and energies. PL intensity changes are interpreted using a theoretical model for reactivity that considers strain along the curvature of a SWCNT. We are also able to estimate the defect density by comparing experimentally obtained quenching with simulations based on numerical solutions of a diffusion equation. Characteristic trapping potential depths of color centers are studied by analyzing emission energies to elucidate the nature of the dopant states. Furthermore, we perform time-resolved PL measurements to investigate the dynamics of color center emission.

## Results and discussion

**PL spectroscopy for individual SWCNTs before and after the functionalization**. Air-suspended SWCNTs are grown across trenches on Si substrates by chemical vapor deposition[31], and vapor-phase reaction using iodobenzene under ultraviolet (UV) irradiation is then conducted to create color centers. Figure 1a shows a schematic of a functionalized nanotube. Scanning electron microscopy confirms that the tubes stay suspended after the functionalization (Fig. 1b). We emphasize that the vapor-phase reaction here differs from typical functionalization techniques established for dispersed SWCNTs in liquid[10–13]. The solution process results in contaminating the tube surface and quenches PL due to interactions between SWCNTs and surrounding environment[32]. It is noteworthy that directly immersing tubes into water inevitably destroys the air-suspended structures due to the high surface tension of the solvent (Supplementary Fig. 1).

We begin by examining PL spectra before and after the functionalization of a (10,5) SWCNT (Fig. 1c) by using the coordinates of the tube on the chip[31] to ensure that we are comparing the same individual tube. The pristine tube only shows exciton emission at a higher energy, whereas the functionalized tube shows two additional peaks at lower energies. We label the exciton emission at 1.02 eV as $E_{11}$ and the additional peaks at 0.94 and 0.88 eV as $E_{11}^-$ and $E_{11}^{-*}$, respectively. The lower energies of $E_{11}^-$ and $E_{11}^{-*}$ indicate that $sp^3$ defects of phenyl group are formed on SWCNTs which introduce dopant states[33]. No remarkable spectral shift of $E_{11}$ emission peak is detected, suggesting negligible changes in dielectric environment due to the vapor residue. The overall PL intensity reduces to less than a quarter which implies introduction of quenching sites in addition to color centers. Hereafter, we refer to defects that decrease the $E_{11}$ intensity as quenching sites and defects that give rise to the $E_{11}^-$ and the $E_{11}^{-*}$ peaks as color centers.

Imaging measurements are performed to characterize the spatial distributions of $E_{11}$, $E_{11}^-$, and $E_{11}^{-*}$ emission. We scan over a (9,7) SWCNT to collect PL spectra and construct intensity maps for the three peaks by spectrally integrating intensities of each peak (Fig. 1d–f). The edges of the trench can be identified using a reflection image in the same area (Fig. 1g). Bright PL is emitted from the suspended region, as typically observed for air-suspended nanotubes[31]. The spatial profile of the $E_{11}$ emission indicates the location of the nanotube and the additional peaks are observed along this tube, consistent with emission originating from color centers formed on the same tube. It is noted that $E_{11}^-$ and $E_{11}^{-*}$ emission show some spatial inhomogeneity in the intensity.

**Statistical investigation of emission intensities**. Emission from various chiralities are now studied by collecting PL spectra of more than 2000 individual SWCNTs before and after the functionalization. All PL data are obtained from a single substrate which assures that the reaction condition is the same, allowing for direct comparison among the chiralities. To acquire the data efficiently, we perform two sets of measurements with excitation energies of 1.46 and 1.59 eV which are near-resonant to many chiralities. Assuming that excitation is close to the $E_{22}$ energy, chiralities of SWCNTs are assigned based on the $E_{11}$ emission energy. For statistical analysis, 12 chiralities with sufficient numbers of tubes are used.

We first consider the intensity of $E_{11}$ emission before the reaction. PL spectra are fitted by a Lorentzian function to obtain the spectrally integrated intensity $I_0$. Large dispersion is observed (Supplementary Figs. 12, 13, and 14), which can be attributed to various suspended lengths (Supplementary Note 1). Many SWCNTs are not fully suspended as observed in PL images (Supplementary Fig. 15), and their suspended lengths are likely shorter than the trench widths. Indeed, the intensity dispersion is well reproduced by simulations of length dependent PL intensity[28,29] assuming a log-normal length distribution[34] centered at 0.78 μm (Supplementary Fig. 16), indicating that most nanotubes have lengths ranging from 0.5 to 1.0 μm. We note that the simulations cannot be directly compared to experiments for nanotubes with no detectable PL, but the fraction of tubes is negligible at low intensities for the distribution reproducing the experimental data.

After the functionalization, color center emission appears in the PL spectra (Fig. 2a, b). All chiralities exhibit $E_{11}^-$ and $E_{11}^{-*}$ emission except for (10, 8), (11, 7), and (12, 5) SWCNTs whose $E_{11}^{-*}$ is beyond the low energy detection limit. We find that most $E_{11}^-$ peaks are taller than $E_{11}^{-*}$ peaks.

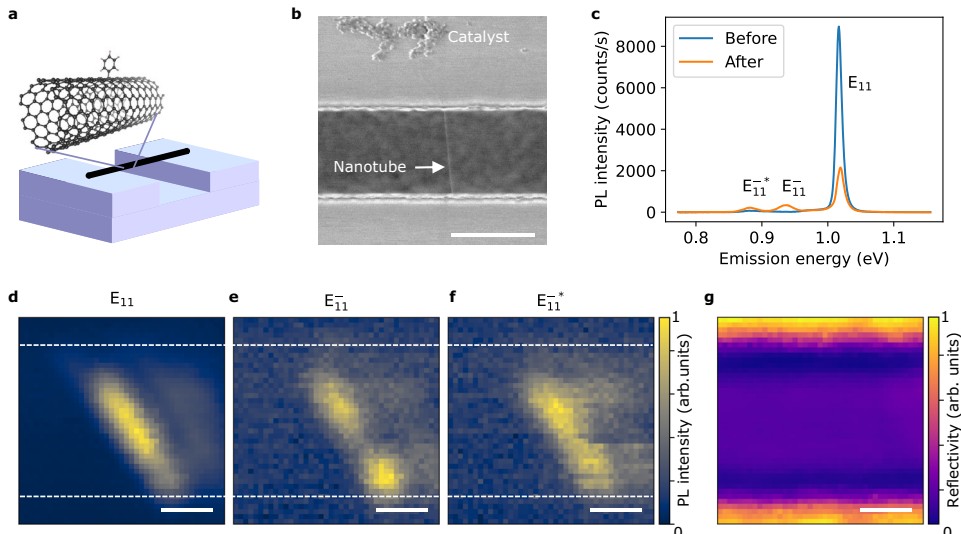

**Fig. 1 Introducing organic color centers to air-suspended nanotubes using vapor-phase reaction. a** A schematic of a functionalized SWCNT suspended across a trench on a Si substrate. **b** A scanning electron micrograph of a tube after the functionalization and the series of PL measurements. Particles on the top are patterned catalysts for growing SWCNTs and the nanotube is indicated by an arrow. **c** Representative PL spectra of an identical air-suspend (10,5) SWCNT before and after the functionalization taken with a laser power of 10 μW and an excitation energy of 1.59 eV. PL intensity maps of (**d**) $E_{11}$, (**e**) $E_{\overline{11}}^-$, and (**f**) $E_{\overline{11}}^{-*}$ emission from a (9,7) tube where the intensity is integrated within a window of 37.4, 32.5, and 28.5 meV centered at each emission peak, respectively. The color scales are normalized to the maximum intensities in the respective maps. The dim features on the right of the tube are caused by reflection of the excitation laser from the bottom of the trench. The white broken lines indicate the edges of the trench. **g** A reflection image in the same area, where brighter and darker regions correspond to the surface of the substrate and the bottom of the trench, respectively. The scale bars in panels (**b**, **d**–**g**) are 1.0 μm. Source data for panels (**b**–**g**) are provided as a Source Data file.

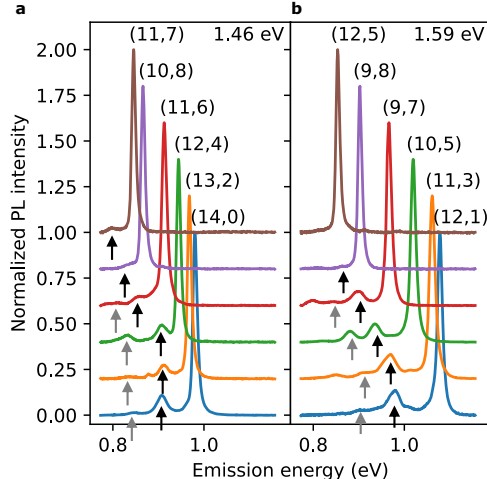

**Fig. 2 Dopant state emission from various chiralities.** PL spectra of functionalized SWCNTs collected with excitation laser energies of (**a**) 1.46 and (**b**) 1.59 eV and with an excitation power of 100 μW, where the spectra are displaced vertically for clarity. Black and gray arrows indicate $E_{\overline{11}}^-$ and $E_{\overline{11}}^{-*}$, respectively. Chirality $(n, m)$ is labeled next to the $E_{11}$ peaks. Source data are provided as a Source Data file. Logscale PL spectra for each chirality are presented in Supplementary Fig. 2.

The intensity of $E_{\overline{11}}^-$ emission can be used to quantify color center density, but care must be taken because of the large variation in the emission intensity. We use the subpeak ratio $I_{\overline{11}}^-/I_{11}$ as a measure of the color center density, where $I_{\overline{11}}^-$ and $I_{11}$ are the spectrally integrated intensities of $E_{\overline{11}}^-$ and $E_{11}$ emission, respectively. By taking the ratio, the variations of $E_{11}$ emission intensity can be canceled out to some degree. The subpeak ratio $I_{\overline{11}}^-/I_{11}$ is plotted as a function of $E_{11}$ emission energy in Fig. 3 and for each chirality in Supplementary Fig. 3. We observe a

monotonically increasing trend with emission energy, indicating that smaller diameter tubes have more color centers (Supplementary Fig. 4). It should be noted that the chirality dependent $E_{22}$ energy (Supplementary Fig. 5) results in emission intensity differences, since we fix the excitation energy either at 1.46 or 1.59 eV. Taking the ratio cancels out the chirality dependent $E_{11}$ emission intensity, allowing for direct comparison between different chiralities. The intensity dispersion can also be caused by resonance shifts due to initial strain generated during growth and inhomogeneity of dielectric environment, but PL intensity variations should be insignificant since energy shifts are typically within ±10 meV[29]. Although small, the effects of these variations are likewise reduced by taking the ratio.

It is also possible to study the effects of the defects from the reduction of $E_{11}$ emission due to the functionalization. In a manner similar to the subpeak ratio, we take the quenching degree $(I_0 - I_{11})/I_0$ as a more physically relevant quantity for comparison between different nanotubes with various suspended lengths and chiralities. The quenching degree exhibits an increasing trend with emission energy as in the case of $I_{\overline{11}}^-/I_{11}$ except for (9, 8) and (12, 5) nanotubes, which do not have a statistically sufficient number of data points (Fig. 3c, d). The large variation of the ratios (Supplementary Fig. 6) are likely caused by multiple factors including inhomogeneity among SWCNTs, temporal fluctuations in intensity (Supplementary Fig. 7), and positions of the defects[35]. We note that $(I_0 - I_{11})/I_0$ reflects the effects of color centers in addition to quenching sites, as both results in reduced $E_{11}$ emission through trapping excitons.

**A model for the chemical reactivity.** To quantitatively interpret the trend of the subpeak ratio and the quenching degree, a theoretical model is developed. Chemical reactivity of SWCNTs depends on both π-orbital pyramidalization angle[36] and π-orbital misalignment angle between adjacent pairs of conjugated C atoms[18,37]. The former is subject to strain arising from the

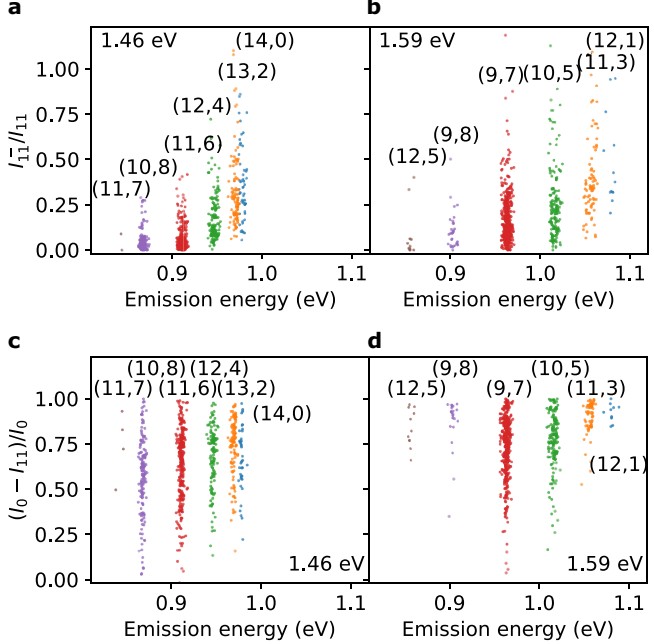

**Fig. 3 Statistical analysis of PL intensities.** Subpeak ratio $I_{11}^-/I_{11}$ as a function of emission energy for experiments conducted with excitation energies of (**a**) 1.46 and (**b**) 1.59 eV and an excitation power of 100 μW. Quenching degree $(I_0 - I_{11})/I_0$ measured with excitation energies of (**c**) 1.46 and (**d**) 1.59 eV and an excitation power of 10 μW. Source data are provided as a Source Data file.

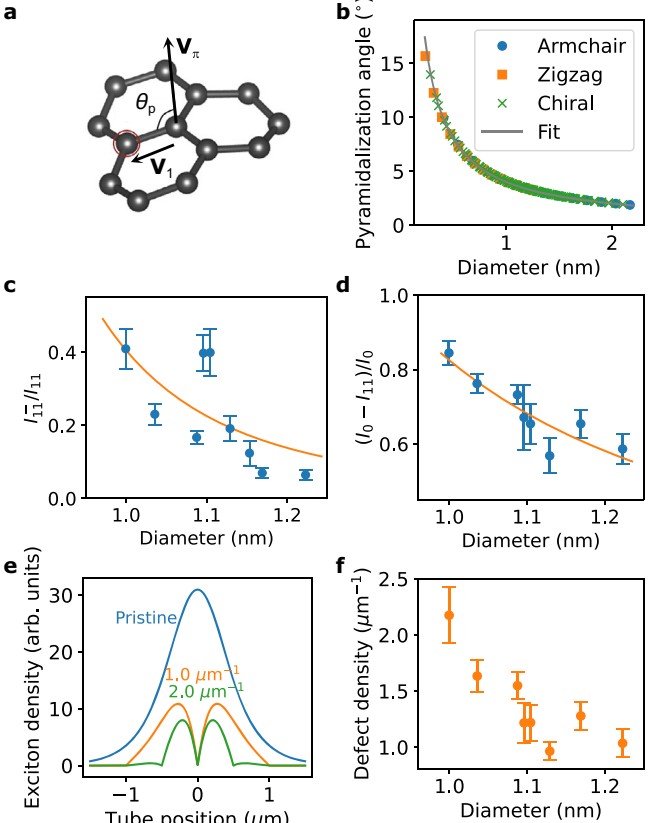

**Fig. 4 Diameter-dependent reactivity. a** A schematic defining the pyramidalization angle $\theta_p$ where $\mathbf{V}_\pi$ is a π-orbital axis vector, $\mathbf{V}_1$ is a unit vector pointing from the target atom to an adjacent atom[55]. The pyramidalization angle can be analytically estimated using a relationship $\cos(\theta_p + 90°) = \mathbf{V}_1 \cdot \mathbf{V}_\pi$. **b** Diameter dependence of computed pyramidalization angle, where the angles of armchair, zigzag, and chiral tubes are separately plotted. Diameter dependence of (**c**) subpeak ratio $I_{11}^-/I_{11}$ and (**d**) quenching degree $(I_0 - I_{11})/I_0$, where error bars are the standard error of the mean. The solid line in panel (**b**) is a fit as explained in the main text, and the solid lines in panels (**c, d**) are fits by Eq. (1). **e** Simulations of $E_{11}$ steady-state exciton density profile for no defects (blue), $\rho = 1.0$ (orange), and $\rho = 2.0$ μm⁻¹ (green). The origin of the coordinate system is taken to be the center of the tube. **f** Diameter dependence of estimated defect density, where error bars are the standard error of the mean. The error bars in (**c, d, f**) are calculated from 5 to 53 measurements for each data point. Source data for panels (**b–f**) are provided as a Source Data file.

curvature of the tubular structure and is diameter dependent, whereas the latter originates from a bond angle with respect to the tube axis and is chiral angle dependent. As we observe a clear diameter dependence, our model considers the π-orbital pyramidalization angle $\theta_p$ depicted in Fig. 4a. The C-C bonds are more bent for larger $\theta_p$, corresponding to larger strain. Figure 4b shows calculated pyramidalization angles as a function of the nanotube diameter $d$ along with a fit by a scaling law $\theta_p = \delta/d$ where $\delta = 4.01°\,\mathrm{nm}^{-1}$ is the coefficient. Because the strain from the curvature increases the chemical reactivity[36], we assume that the reduction in the activation energy is proportional to the pyramidalization angle. The activation energy of the reaction is then $E_a(d) = E_a(\infty) - \eta\theta_p(d)$ where $E_a(\infty)$ is the activation energy for graphene, and $\eta$ is the coefficient. According to the Arrhenius equation, it follows that the chemical reaction rates are proportional to

$$\exp\left(-\frac{E_a(\infty)}{k_B T} + \frac{1}{k_B T} \cdot \frac{\eta\delta}{d}\right) \qquad (1)$$

where $k_B$ is the Boltzmann constant and $T = 298$ K is the temperature. The ratios $I_{11}^-/I_{11}$ and $(I_0 - I_{11})/I_0$ should therefore scale as $\exp\left(\frac{1}{k_B T} \cdot \frac{\eta\delta}{d}\right)$.

Since data dispersion is large, we use the average values of these ratios for each tube diameter (Fig. 4c, d). We only include the tubes on the widest trenches with 3.0 μm width to reduce end quenching effects. It is confirmed that SWCNTs with smaller diameters show higher ratios, indicating that these tubes are more reactive despite the smaller surface areas. We fit the model to the experimental data, and both ratios show good agreement. The fit to $I_{11}^-/I_{11}$ yields $\eta = 41.3 \pm 9.4$ meV/deg, whereas the fit to $(I_0 - I_{11})/I_0$ results in $\eta = 13.4 \pm 1.9$ meV/deg. The higher $\eta$ for $I_{11}^-/I_{11}$ by a factor of 3.08 indicates that the formation of color centers is more responsive to strain compared to quenching sites, although the higher excitation power used to obtain the subpeak

ratio could lead to underestimation of the reactivity for the color centers (Supplementary Fig. 18 and Note 2). It should be possible to use this difference for preferential creation of color centers by choosing tubes with smaller diameters. We note that an opposite diameter dependence on chemical reactivity has been reported for a reaction with 4-hydroxybenzene diazonium[38], where electron transfer limits the reaction rate.

**Defect density estimation.** The ratio $(I_0 - I_{11})/I_0$ also allows us to quantify the defect density by modeling the effects of functionalization on PL intensity. We use the fact that pristine air-suspended nanotubes are defect free except for end quenching[28,29]. Although the nanotubes used in this study have varying suspended lengths (Supplementary Note 1), the experimentally obtained distribution of $I_0$ (Supplementary Fig. 12) is consistent with a simulation assuming defect free nanotubes (Supplementary Fig. 16). In comparison, the experimental results

cannot be reproduced if initial defects are included in the simulations (Supplementary Fig. 17), and we therefore estimate the defect density in pristine SWCNTs to be much less than 0.25 μm$^{-1}$.

Our model is based on a steady-state one-dimensional diffusion equation

$$D\frac{d^2 n(z)}{dz^2} - \frac{n(z)}{\tau} + \frac{\Gamma_0}{\sqrt{2\pi r^2}}\exp\left(-\frac{z^2}{2r^2}\right) = 0 \quad (2)$$

where $D$ is the diffusion coefficient, $n(z)$ is the $E_{11}$ exciton density, $z$ is the position on the tube, $\tau = 70$ ps is the intrinsic lifetime of excitons[31], $\Gamma_0$ is the exciton generation rate, and $r = 530$ nm is the $1/e^2$ radius of the laser spot. The first term accounts for the exciton diffusion, the second term represents the intrinsic recombination, and the third term describes exciton generation which is proportional to the Gaussian laser profile. We consider SWCNTs with infinite length and set the boundary conditions to be $n(\pm\infty) = 0$. Additional boundary conditions $n(z_d) = 0$ are imposed for the functionalized tube where $z_d$ is the position of defects, assuming that the sites are uniformly distributed with density $\rho$. For the diffusion coefficient, we use the expression $D = D_0(d/d_0)^\alpha$ where $D_0 = 15.36$ cm$^2$/s is the diffusion coefficient at diameter $d_0 = 1.00$ nm corresponding to a diffusion length $\sqrt{D\tau} = 328$ nm, and $\alpha = 2.56$ is the exponent[31]. The diffusion equation is numerically solved to obtain $n(z)$ and the results are plotted for various site densities in Fig. 4e (Supplementary Fig. 8a). When the defect separation is much shorter than the laser spot diameter, the quenching process becomes dominant over the intrinsic decay and results in a significant decrease in the exciton density.

The defect density is estimated by comparing $(I_0 - I_{11})/I_0$ obtained from the experiments with the simulations. We use the experimental data of tubes on the widest trenches with 3.0 μm widths, and the PL intensity in the simulation is computed by integrating $n(z)$ (Supplementary Fig. 8b). The only unknown parameter $\rho$ is extracted by matching the simulated quenching degree with the experimental values. The results are plotted as a function of the diameter in Fig. 4f. The defect density ranges from 0.95 to 2.2 μm$^{-1}$ and shows a diameter dependence which is consistent with the trend of the reactivity (Fig. 4c, d). The estimated defect density represents a lower bound because of the assumption in the simulations that defects are uniformly distributed in nanotubes (Supplementary Note 3). We also compare the distribution of $(I_0 - I_{11})/I_0$ between experiments and simulations (Supplementary Note 4). The experiments are well reproduced by a simulation assuming the log-normal length distribution and $\rho = 1.5$ μm$^{-1}$ (Supplementary Figs. 19 and 20a). In comparison, a simulation with a high defect density of $\rho = 10$ μm$^{-1}$ cannot reproduce the experimental results (Supplementary Fig. 20b). It is worth mentioning that the estimated defect density includes contributions from color centers and quenching sites as they both reduce the number of $E_{11}$ excitons.

**Emission energy analyses for various chiralities.** We now proceed to analyze $E_{11}^-$ and $E_{11}^{-*}$ peak positions. Energy separations $\Delta E_{11}^- = E_{11} - E_{11}^-$ and $\Delta E_{11}^{-*} = E_{11} - E_{11}^{-*}$ can be interpreted as trapping potential depths for $E_{11}^-$ and $E_{11}^{-*}$ excitons, respectively, and we plot $\Delta E_{11}^-$ and $\Delta E_{11}^{-*}$ as a function of $E_{11}$ in Fig. 5a, b. The energy separations show correlated increase with $E_{11}$, confirming that $E_{11}^-$ and $E_{11}^{-*}$ originate from dopant states of $E_{11}$ exciton, and not of other states such as $E_{22}$ and $E_{33}$ excitons. It is noted that we observe smaller clusters for $\Delta E_{11}^-$ as marked by circles in Fig. 5a, whose trapping potentials are smaller than the main clusters. The differences in $E_{11}^-$ could be assigned to different binding configurations of the phenyl functional group, where ortho- and para-configurations exhibit different emission energies[11,18,39]. We

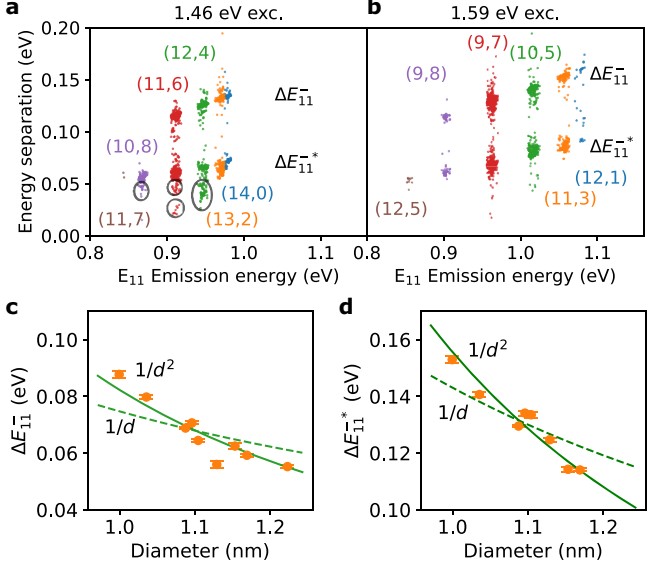

**Fig. 5 Statistical analysis of emission energies.** Energy separation $\Delta E_{11}^{-*}$ and $\Delta E_{11}^-$ of functionalized SWCNTs as a function of emission energy, where the data are collected with excitation energies of (**a**) 1.46 and (**b**) 1.59 eV and a power of 100 μW. The color of the dots represents the nanotube chirality and isolated clusters are indicated by circles. Diameter dependence of average (**c**) $\Delta E_{11}^-$ and (**d**) $\Delta E_{11}^{-*}$, where error bars are the standard error of the mean and calculated from 25 to 373 measurements for each data point. The solid and broken lines are fits by the power laws $1/d^2$ and $1/d$, respectively. The data are better described by $\Delta E_{11}^- = A^-/d^2$ and $\Delta E_{11}^{-*} = A^{-*}/d^2$, where $A^- = 84.0 \pm 2.64$ meV·nm$^2$ and $A^{-*} = 154 \pm 1.43$ meV·nm$^2$ are the coefficients for $\Delta E_{11}^-$ and $\Delta E_{11}^{-*}$, respectively. Source data are provided as a Source Data file.

similarly interpret the $E_{11}^{-*}$ emission to be arising from other binding configurations.

The diameter dependence of the trapping potentials provides additional insight to the nature of the dopant states. The average values of $\Delta E_{11}^-$ and $\Delta E_{11}^{-*}$ for each chirality are plotted as a function of the diameter in Fig. 5c, d. The dependence shows a monotonic decrease and differs from the constant energy separation of 130 meV for the $K$-momentum excitons, and we thus exclude them from the origin of $E_{11}^-$ and $E_{11}^{-*}$. To describe the dependence, we consider the $1/d$ scaling observed for the exciton binding energies and the $1/d^2$ scaling for singlet-triplet splitting[40–44] (Supplementary Note 5). The $1/d^2$ scaling yields better fits to both $\Delta E_{11}^-$ and $\Delta E_{11}^{-*}$ than the $1/d$ scaling.

The values of the energy separations $\Delta E_{11}^-$ and $\Delta E_{11}^{-*}$ observed in this work show more similarity to triplet[42,43] and trion states[45]. Examining the higher energy peak $E_{11}^-$, $\Delta E_{11}^- = 84.0$ meV at $d = 1$ nm is as high as the energy splitting reported for triplet excitons; Matsunaga et al. found $1/d^2$ dependence of energy separation for laser-induced defects in air-suspended SWCNTs[43], with a value of 70 meV at $d = 1$ nm. Nagatsu et al. showed the $1/d^2$-dependent energy separation in air-suspended SWCNTs for H$_2$-adsorption-induced peaks where $\Delta E_{11}^- = 68$ meV at $d = 1$ nm is attributed to triplet excitons[42]. The comparable values of the energies and the $1/d^2$ scaling for $E_{11}^-$ suggest that the triplet exciton state is brightened at the color centers. Further study with optically detected magnetic resonance[46] and magneto-PL spectroscopy[47] would be required to clarify the triplet origin of $E_{11}^-$ excitons. Considering the lower energy peak $E_{11}^{-*}$, $\Delta E_{11}^{-*} = 154$ meV at $d = 1$ nm is close to 175 meV-separation between exciton and trion energies for air-suspended SWCNTs[45] with a diameter of 1 nm. Using chemical[47] or electric-field[48] doping to investigate

trions trapped at color centers in air-suspended nanotubes may help elucidate the origin of the state. We note that the tubes studied here have larger diameters than typical SWCNTs dispersed in liquid[11,18], but the observed $\Delta E_{11}^-$ and $\Delta E_{11}^{-*}$ are consistent with the extrapolation of the $d$ dependence for smaller diameters.

The diameter dependence of the trapping potential is partially responsible for the difference of subpeak ratio between air-suspended and solution-processed nanotubes[33]. The larger diameters for air-suspended SWCNTs lead to shallower trapping potentials[15,22], resulting in lower quantum yields for $E_{11}^-$ emission. In addition, pristine air-suspended nanotubes can be considered defect free[28,29], whereas solution-processed tubes have naturally formed defects with a typical density of 8.3 μm$^{-1}$ [30]. $I_0$ in air-suspended nanotubes is therefore stronger than solution-processed tubes, further reducing the subpeak ratio.

**The dynamics of excitons in a functionalized SWCNT.** The dynamics of the color center emission is also investigated by time-resolved PL measurements to compare to the decay lifetimes of $E_{11}$. We use a pulsed laser for excitation, and the $E_{11}$ emission is spectrally differentiated from the $E_{11}^-$ and $E_{11}^{-*}$ emission with a band-pass filter and a long-pass filter (Supplementary Fig. 9). Figure 6 shows PL decay curves for these emission peaks from an (11, 3) functionalized tube. Decay lifetimes are extracted by fitting a biexponential function $a_1 \exp(-t/\tau_1) + a_2 \exp(-t/\tau_2)$ convoluted with the instrument response function, where $\tau$ is the decay lifetime and $a$ is the amplitude with the subscripts 1 and 2 denoting fast and slow components, respectively. We find that $\tau_1$ for color center emission is 1.6 times longer than $\tau_1$ for $E_{11}$ exciton emission. This difference is moderate compared to 3.3 times observed for solution-processed SWCNTs[15] likely because of the lower defect density in air-suspended nanotubes resulting in longer $E_{11}$ exciton lifetime.

The biexponential behavior for $E_{11}$ and $E_{11}^-$ emission is consistent with previous reports for solution-processed samples[15,22,49], where the decay for $E_{11}$ can be understood by a three-level model including bright, dark, and ground states[50,51]. Both bright and dark excitons are populated by $E_{22}$ resonant excitation and inter-state transitions take place between the bright and dark states, resulting in the fast component from bright

exciton recombination and the slow component reflecting the dynamics of dark excitons. The bixponential decay for color center emission can be similarly understood by considering an $E_{11}^-$ manifold having three levels associated with the color center which is independent from the $E_{11}$ states[15,23].

The lifetimes $\tau_1 = 69.1$ and $\tau_2 = 172.0$ ps for the color centers of the air-suspended nanotube are comparable to $\tau_1 = 86$ and $\tau_2 = 171$ ps for solution-processed tubes[15]. This result is reasonable because color centers are isolated and other defects should not affect the excitons trapped at the color centers. The comparable lifetimes also indicate that the vapor-phase reaction can form as high-quality color centers as the solution-processed methods. Although the diameter dependence of the potential depth for $E_{11}^-$ exciton suggests its triplet nature, the observed lifetime is several orders of magnitude shorter than 30–200 μs reported for the triplet states[46,52]. Lifetime shortening may be caused by hybridization of the singlet and triplet excitons[53].

Finally, we discuss how improvements can be made to take full advantage of the vapor-phase reaction. The formation of color center density can be adjusted by optimizing the duration of reaction time (Supplementary Fig. 10), ideally creating a single color-center per nanotube. We are aware of quenching sites introduced by UV irradiation itself, as observed in control experiments in the absence of iodobenzene (Supplementary Fig. 11). Suppression of this process could be possible by optimizing the power and the energy of UV light. By introducing deeper traps by different precursor molecules, quantum yield for $E_{11}^-$ may be improved[22,33]. It is desirable to develop vapor-phase chemistry with these molecules for air-suspended nanotubes as the quantum yield in solution-processed nanotubes has improved by ~50% with a trapping potential deeper by 16 meV[22]. Because the potential depth depends on the diameter (Fig. 5c, d), SWCNTs with smaller diameters would be suited for higher quantum yield.

In summary, we have demonstrated functionalization of air-suspended SWCNTs using iodobenzene as a precursor. The comparison of PL spectra before and after the vapor-phase reaction shows additional peaks $E_{11}^-$ and $E_{11}^{-*}$ from color centers and PL intensity reduction of the $E_{11}$ peaks. Twelve representative chiralities are characterized using spectra from more than 2000 individual tubes, where the diameter dependent subpeak ratio and quenching degree are observed. We have modeled the diameter dependent reactivity which is found to be proportional to $\exp(1/d)$, explaining the experimental results. By further performing the exciton diffusion simulations, we have estimated the defect density and found that these values are also diameter dependent. The analysis of peak energies reveals that both $E_{11}^-$ and $E_{11}^{-*}$ states originate from dopant states of $E_{11}$ excitons and have trapping potentials scaling as $1/d^2$. We observe a longer PL lifetime for dopant states, similar to the reports on solution-processed tubes. By elucidating the exciton physics as well as functionalization chemistry, color centers in air-suspended SWCNTs should provide new opportunities in photonics and optoelectronics for quantum technologies.

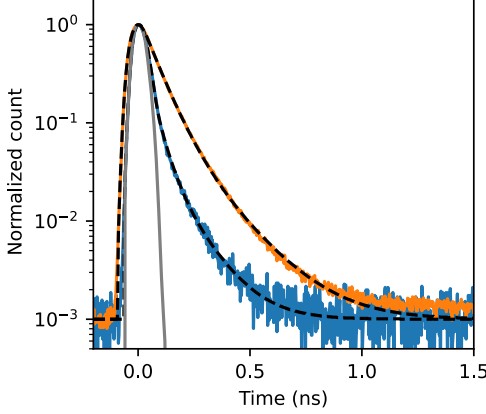

**Fig. 6 Time-resolved PL properties.** PL decay curves of $E_{11}$ emission (blue) and $E_{11}^-$ and $E_{11}^{-*}$ emission (orange) from a functionalized (11, 3) SWCNT suspended across a trench with a width of 1.0 μm measured by an excitation energy of 1.59 eV and a power of 10 nW using the pulsed laser. The broken lines are the fits by a biexponential decay function convoluted with the instrument response function (gray), showing $\tau_1 = 42.2$ and $\tau_2 = 122.0$ ps for $E_{11}$ and $\tau_1 = 69.1$ and $\tau_2 = 172.0$ ps for $E_{11}^-$ and $E_{11}^{-*}$. Source data are provided as a Source Data file.

## Methods

**Air-suspended carbon nanotubes.** Electron-beam lithography and dry etching are used to fabricate trenches on Si substrates[54] with a depth of ~1 μm and a width of up to 3.0 μm. Another electron-beam lithography is conducted to define catalyst areas near trenches, and Fe-silica catalyst dispersed in ethanol are spin-coated and lifted off. SWCNTs are synthesized over trenches using alcohol chemical vapor deposition[31,54] under a flow of ethanol with a carrier gas of Ar/H$_2$ at 800 °C for 1 min.

**Formation of organic color centers.** Vapor-phase reaction is used to functionalize air-suspended nanotubes with iodobenzene as a precursor. As-grown SWCNTs on the Si substrates are placed facing up inside a glass chamber having a diameter of

15 mm and a height of 5 mm. Iodobenzene (5 µL) is introduced to the bottom of the chamber by a micropipette. The chamber is then covered with a quartz slide and sealed using high vacuum grease. We leave the chamber for 10 min to fill it with iodobenzene vapor, after which the reaction is triggered by irradiating the sample with 4.88-eV UV light through the quartz slide. The samples are collected from the chamber and stored in dark for characterization by subsequent spectroscopy.

**Micro-PL measurements**. PL spectra are obtained with a home-built scanning confocal microscope[31], where we use a continuous-wave Ti:sapphire laser for excitation and a liquid-$N_2$-cooled InGaAs photodiode array attached to a 30-cm spectrometer for detection. Laser polarization is kept perpendicular to trenches, and the beam is focused using an objective lens with a numerical aperture of 0.85 and a focal length of 1.8 mm. The $1/e^2$ diameters of the focused beams are 1.31 and 1.06 µm for excitation energies of 1.46 and 1.59 eV, respectively, where the diameters are characterized by performing PL line scans perpendicular to a suspended tube. PL excitation spectroscopy is conducted by scanning excitation wavelength at a constant power[29]. The reflected laser light is collected with a biased Si photodiode for reflection images. All PL spectra and decay curves are taken at the center of the nanotubes except for the hyperspectral PL imaging.

**Spectral analysis**. Peak parameters are extracted by fitting a Lorentzian function to PL spectra for pristine SWCNTs and a triple-Lorentzian function for functionalized SWCNTs. Pristine nanotubes with an $E_{11}$ peak height of less than 500 counts/s are excluded in the statistics and not used for further measurements. We also exclude functionalized nanotubes with an $E_{11}$ peak height of less than 400 counts/s.

**Decay lifetime measurements**. For time-resolved PL measurements, the Ti:sapphire laser is switched from continuous wave to ~100-fs pulses with a repetition rate of 76 MHz. A fiber-coupled superconducting single-photon detector is used to measure PL decay. Emission from $E_{11}$ excitons and dopant states are separately obtained with a band pass filter and a long pass filter.

## Data availability

All the data generated in this study have been deposited in the R2DMS-GakuNinRDM database at https://dmsgrdm.riken.jp/nxuar/. Source data are provided with this paper.

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

## Acknowledgements

This work is supported in part by MIC (SCOPE 191503001 to Y.K.K.), JSPS (JP18H05329, JP20H00220 to S.M., JP19H02543, JP20KK0114 to R.X., JP20H02558 to Y.K.K., JP20K15112 to D.K., JP20K15137 to K.O.), MEXT (Nanotechnology Platform JPMXP09F19UT0077), NSF (RAISE-TAQS PHY-1839165 to Y.H.W.), and JST (CREST JPMJCR20B5 to S.M.). D.K. acknowledges support from RIKEN Special Postdoctoral Researcher Program. K.O. is supported by JSPS Research Fellowship. We thank the Advanced Manufacturing Support Team at RIKEN for technical assistance.

## Author contributions

Y.K.K., Y.H.W., and S.M. conceptualized the idea, and formulated the overarching research goals. A.I., T.I., and R.X. synthesized SWCNTs. X.W. and J.F. conducted the functionalization. D.K., A.I., and K.O. performed the optical measurements. D.K., A.I., and Y.K.K. interpreted the results. D.K. and Y.K.K. wrote the original draft with input from all the authors.

## Competing interests

The authors declare no competing interests.
