## [Peer Review File · Nature Communications]

REVIEWER COMMENTS

Reviewer #1 (Remarks to the Author):

In this paper, D. Kozawa et al present a new way of synthesizing organic color centers on air-suspended SWCNTs. While such color center synthesis is very well known for SWCNTs dispersed in solutions, a comparable synthesis strategy for air-suspended SWCNTs was not yet available. The authors clearly demonstrate the functionalisation of their SWCNTs and provide conclusive evidence that the functionalisation results in both defects (quenching sites) as optical traps for the mobile excitons. The authors also show that the observed diameter-dependent functionalisation can be easily modelled by incorporating the strain on the C-bonds in the SWCNT surface. The results are novel, well-supported, and the number of statistics is impressive. Therefore, I certainly recommend publication in a high-ranked journal like Nature Communications.

Before publication, I do have a couple of small remarks that the authors should take into account.

1) In Figure 1d-f, the authors show a PL-intensity spatial map, selecting either the emission from the E11 or E11- states. This shows that the distribution of the defect sites changes drastically along the length of a nanotube.

a) I wonder if the PL intensity scale in the 3 panels is the same (which would indicate that at some positions the E11- is stronger than the E11 peak?)  is this the integrated or peak intensity which is plotted?

b) When plotting the spectrum of a single tube (panel c for example), is this then the spectrum at one specific location or is it an average of all spectra in the spatial map?

c) Why do the E11- and E11-* show intensity on the right of the CNT (i.e. when making e.g. a horizontal profile in these PL spatial maps, the E11 will be very narrow, but the E11- and E11-* will have an intensity outside of the SWCNT position? How can this be explained if the defects are functionalised on the SWCNT?

2) On page 4 the authors make the remark: "it is noteworthy that directly immersing air-suspended tubes into water inevitably destroys the structures due to"

 Do they mean the defect sites? or the SWCNTs themselves?

3) It would be good if they could indicate which peaks they assign to E11- and E11-* in Figure 2, as some of the spectra seem to show more than 2 side-bands in the PL spectra? Perhaps indicate this with arrows in each spectrum?

4) For the statistical data, I was wondering if they used each SWCNT only once in the data set (so measuring on 2000 separate SWCNTs?). If yes, which intensity was then used, since a single tube seems to emit so inhomogeneously (see figure 1).

5) Did the authors try to change the reaction time, to enhance/decrease the number of defects on the SWCNTs? This would in particular be interesting as in theory one would probably prefer only a very limited number of color centres, and no defects that quench the emission. If not tested, the authors should definitely comment on this in the future outlook if they think this would be feasible.

6) When the authors refer to the triplet states, I would recommend them to also compare their results with this paper: J. <https://doi.org/10.1021/acsnano.0c03139>, which demonstrates the singlet-to-triplet energy difference for multiple chiralities.

7) The authors mention that they need to do the synthesis under UV illumination. Did they also test if the UV illumination itself could not cause such defects (thinking of the original UV - based doping work: DOI: 10.1126/science.1196382)

8) Several of the research works that the authors refer to regarding triplet excitons, show lifetimes in the microsecond regime, while here the authors see only a minor change in the lifetimes. Also for organic color centers synthesized for SWCNTs in dispersion, typically a much larger effect on the lifetimes is observed. Could the authors comment on this? Does this have to do with the large number of quenching sites present?

9) In figure 4c, could the authors also plot the spread on the data points (instead of or together with the error of the mean?). I think the spread is much more important. Furthermore, looking at figure 3, it would be interesting to provide a histogram plot of those distributions (perhaps in the SI) which would show that the distribution is quite asymmetric.

Reviewer #2 (Remarks to the Author):

The authors present a study on color-center and defect formation on suspended single wall carbon nanotubes through a gas phase doping process. Although the experiment is likely challenging and thus the large data set covering many chiral angles and diameters is impressive the extractable information seems limited by experimental choices. The most prominent achievements are the distinguishing of reactivity differences and mechanism winnowing. Some information may simply be missing due to the brevity of a communication template. I would support publication as a communication if the following items are addressed.

significant questions:

1. The primary comment is that this study appears to probe a relatively high level of functionalization rather than the sparse functionalization regime. The authors argue the opposite with the presented diffusion model, but I am concerned that the model a, does not consider the presence of some number of defects independent of functionalization or b, the potential for annihilation from the trion formation mechanism proposed later. Is the distribution of IO intensities sufficiently monomodal to argue that the sparse regime is correct?
2. Relatedly, the emission from induce color centers in an absolute sense should be parabolic-like, increasing from no functionalization to a maximum above which the increased number of defects leads to greater quenching. With the observed reactivity differences are all nanotube (n,m) structures observed in the same limit? Can additional information be decoded from the distribution of observed spectra? Publication of the data set of pre and post functionalized observations could enable other researchers to explore such features if the authors argue that it is beyond the scope of this work.
3. Can the authors comment on how spatial inhomogeneity of functionalization may affect their observations and conclusions? Are functionalization sites close to the trench ends observable? Or do these contribute a region of color centers observed as defects? The authors note that different trench widths were utilized but do not discuss this parameter.
4. Experimentally the authors trigger functionalization by a UV source, but other works induce functionalization by E22 excitation, are the spectra static and repeatable when a tube is observed several independent times?
5. Spectra in Figure 1C are taken at a power of 10 uW, but most data is reported for 100 uW excitation. The observation of trion and triplet phenomenon could be tied to the power of observation. Is the result independent of power? Was the experiment limited by practicality concerns to such conditions? Addition of short discussion is in order.

Errata:

Figure S4 the E11-* label is placed over a line feature that is very close to 1600 cm⁻¹ from the E11 emission. The vastly narrower linewidth might also imply that it is a Gband feature.

Many of the emission line scans in Figure S2 appear to show 3 to 4 features. Are these attributable to additional defect configurations? Raman features?

Figure S5B: density is misspelled on the x-axis.

Reviewer #3 (Remarks to the Author):

In the manuscript, "Formation of Organic Color Centers in Air-Suspended Carbon Nanotubes Using Vapor-Phase Reaction," the authors introduce a new functionalization technique that adds organic color centers to the single-walled carbon nanotube (SWCNT) sidewall. Covalent functionalization of air-suspended SWCNTs occurs via a photochemical reaction with adsorbing precursor molecules, which the authors say is desirable to the standard solution-phase chemistry that can contaminate the tube surface and alter photoluminescence (PL) characteristics due to interactions between the nanotube (NT) and local environment. Observations from roughly 2000 individual SWCNTs of varying chiralities revealed the formation of two defect-related emission peaks, referred to as E11- and E11-*, that appear at lower energies from the E11. In conjunction with experimental data, the authors developed theoretical models and simulations aimed at uncovering the nature of the defect states. Time-resolved PL measurements were also acquired to investigate the dynamics of the defect states.

The concepts discussed in the manuscript have been of significant interest to the NT community for some time. Indeed, in the last few years several studies have demonstrated 1) the unique single photon emitting properties of quantum defect-decorated SWCNTs, 2) ways to circumvent the creation of multiple photoactive defect states and 3) approaches to minimize perturbations to the integrity of the NT system due to interactions with the local environment. This manuscript contributes to the same overall body of research in this topical area, but besides providing an alternative functionalization method for creation of color-centers, it is not clear that it has the high novelty and breadth of impact expected of publications in Nature Communications. Indeed, the main conclusions that were drawn from theoretical modeling and simulations are not new or unique when compared to previous studies of defect-decorated SWCNTs: diameter-dependent defect emission shifts have been observed and calculations of defect density have been performed. The manuscript, although interesting, largely presents itself as simply another way to covalently form quantum defect sites on SWCNTs. Therefore, the scope seems to be focused on the narrow range of scientists who are interested in nanotube defect photophysics.

In addition, the experimental data included in this manuscript is often presented without appropriate discussion or explanation, leaving the reader with many fundamental questions (see comments below). The manuscript would benefit from a paragraph or two that clearly describes why this work is important, what these results mean for the NT community, and how improvements can be made to realize the full potential of the vapor-phase reaction method.

Comments:

1a) The E11- and E11-* emission features appear significantly weaker than the E11 emission feature (Figures 1 and 2). Previous studies of quantum defect-decorated SWCNTs have shown that defect emission features will dominate the SWCNT PL spectrum because excitons are rapidly funneled into the defect states. The authors do not address this discrepancy directly in the manuscript. Why do the vast majority of E11- and E11-* features appear so weak?

1b) Are the relatively weak features of E11- and E11-* a consistent shortcoming of the vapor-phase reaction method?

1c) What do the relatively weak features of E11- and E11-* (compared to E11) imply about the underlying photophysics of the exciton in the suspended NTs (versus the solution phase NTs)?

1d) Can the vapor-phase reaction method be competitive with established NT sidewall functionalization techniques if it consistently yields weak defect emission features?

A detailed discussion of these topics is needed to improve the manuscript.

2a) On page 5, the authors note that spatial overlap between E11 emission and defect emission is "expected." The authors need to explain and clarify why they think so? Organic color centers should appear more localized than typical E11 emission (even after accounting for diffraction limited spots).

2b) Is it generally true that there appears to be more PL localization for E11- emission than for E11-* emission (refer to Figures 1e and 1f)?

3) PL quenching seems to be an issue with the vapor-phase reaction method. Can the authors comment on the ability to control whether a defect created via the vapor-phase reaction method becomes an organic color center or quenching site?

3a) There is a brief discussion on page 8 that states, "...the formation of color centers is more responsive to strain compared to quenching sites." This is a very interesting statement, but the authors do not expand on it. A more detailed discussion as to why some defects become color centers and other defects become quenching sites would seem to be required.

4) It is unclear how the scattered data points in Figure 3 were extracted from the PL images. Was this an average value? Peak value? How are PL intensity variations across the NT accounted for?

5) Figures 5a and 5b appear very busy and the trends are difficult to follow, largely due to the chirality labels on the plot. Remaking the figure such that a color-coded key is included to identify the chiralities would be helpful.

6) Did the authors attempt to collect separate time decays for the E11- and E11-* features? Insight into the potential similarities or differences in PL dynamics for each defect feature could be very meaningful.

7) The single paragraph discussing the PL lifetimes of the E11- and E11-* defect states lacks depth. There is no discussion of PL dynamics beyond simply stating the fitting parameters and time constants. On page 11, the authors state: "The fast component is assigned to the decay of bright excitons whereas the slow component reflects the dynamics of dark excitons." While this statement is true, what does this mean with regards to the organic color centers. How are bright and dark excitons interacting with these defect sites? It might be too early to provide an exact explanation, but a discussion on the potential dynamics would be useful.

8) The use of the label "organic color centers" often is suggestive of single-photon characteristics. Do the authors have any evidence that these defect-decorated air-suspended SWCNTs are in fact single photon sources at room temperature? Or is the use of "organic color centers" simply based on the similarity to established work?

Response to reviewer report for manuscript NCOMMS-21-07736A by Kozawa *et al.*

We thank the reviewers for taking their time to review the manuscript, and we are delighted that the three reviewers have found our results to be *novel, impressive, and interesting*. We also thank the reviewers for their helpful comments, and we have used them to improve our manuscript. Our point-by-point response is provided below.

Response to Reviewer #1

In this paper, D. Kozawa et al present a new way of synthesizing organic color centers on air-suspended SWCNTs. While such color center synthesis is very well known for SWCNTs dispersed in solutions, a comparable synthesis strategy for air-suspended SWCNTs was not yet available. The authors clearly demonstrate the functionalisation of their SWCNTs and provide conclusive evidence that the functionalisation results in both defects (quenching sites) as optical traps for the mobile excitons. The authors also show that the observed diameter-dependent functionalisation can be easily modelled by incorporating the strain on the C-bonds in the SWCNT surface. The results are novel, well-supported, and the number of statistics is impressive. Therefore, I certainly recommend publication in a high-ranked journal like Nature Communications. Before publication, I do have a couple of small remarks that the authors should take into account.

We thank the reviewer for the positive evaluation of our work with remarks “*novel, well supported*” and “*impressive*”.

1) In Figure 1d-f, the authors show a PL-intensity spatial map, selecting either the emission from the E11 or E11- states. This shows that the distribution of the defect sites changes drastically along the length of a nanotube. a) I wonder if the PL intensity scale in the 3 panels is the same (which would indicate that at some positions the E11- is stronger than the E11 peak?)  is this the integrated or peak intensity which is plotted?

We thank the reviewer for pointing this out. This description was missing in the original manuscript. The PL intensity scales in the 3 panels are different and the color scales are normalized to the maximum intensities in the respective maps. We have added the description to the caption of Fig. 1d-1f in the revised manuscript. The intensity for E₁₁ is the strongest as shown in the PL spectrum (Fig. 1c), and we plot spectrally integrated intensity as written in the caption of Fig. 1d-1f.

b) When plotting the spectrum of a single tube (panel c for example), is this then the spectrum at one specific location or is it an average of all spectra in the spatial map?

We thank the reviewer again for pointing out a missing detail. The spectra were taken at the center of nanotubes. We have accordingly added this description on line 300 of page 15 in the revised manuscript.

c) Why do the E11- and E11- show intensity on the right of the CNT (i.e. when making e.g. a horizontal profile in these PL spatial maps, the E11 will be very narrow, but the E11- and E11-* will have an intensity outside of the SWCNT position? How can this be explained if the defects are functionalised on the SWCNT?*

The dim features on the right of the tube are caused by reflection of the excitation laser from the bottom of the trench. Since the PL intensities for E_{11}^- and E_{11}^{-*} are relatively low, the signals are close to the baselines of the PL images, resulting in the duplicates. This artifact can be also seen in E_{11} . We have added the explanation to the caption of Fig. 1 in the revised manuscript.

2) On page 4 the authors make the remark: "it is noteworthy that directly immersing air-suspended tubes into water inevitably destroys the structures due to" Do they mean the defect sites? or the SWCNTs themselves?

We mean that the air-suspended structures are destroyed. We have accordingly revised the sentence on line 79 of page 4 in the revised manuscript.

3) It would be good if they could indicate which peaks they assign to E11- and E11- in Figure 2, as some of the spectra seem to show more than 2 side-bands in the PL spectra? Perhaps indicate this with arrows in each spectrum?*

As the reviewer suggested, we have added black and gray arrows that indicate peak positions of E_{11}^- and E_{11}^{-*} emission, respectively, to Fig. 2 in the revised manuscript.

4) For the statistical data, I was wondering if they used each SWCNT only once in the data set (so measuring on 2000 separate SWCNTs?). If yes, which intensity was then used, since a single tube seems to emit so inhomogeneously (see figure 1).

Yes, we used each SWCNT only once in the data set measuring 2099 separate SWCNTs, in which intensity at the center of tubes was used, as clarified in the response to the comment 1b.

5) Did the authors try to change the reaction time, to enhance/decrease the number of defects on the SWCNTs? This would in particular be interesting as in theory one would probably prefer only a very limited number of color centres, and no defects that quench the emission. If not tested, the authors should definitely comment on this in the future outlook if they think this would be feasible.

Yes, we have tested other reaction time and found that longer reaction time results in losing almost the entire fluorescence signals due to formation of high-density quenching sites (Supplementary Fig. 12). We agree with the reviewer that the optimization of the reaction time is important and plan to systematically study this problem in a future work. We have added these discussions on line 253 of page 13 and Supplementary Fig. 11-12 in the revised manuscript. Please also see our response to Reviewer #3's Comment 1d.

6) When the authors refer to the triplet states, I would recommend them to also compare their results with this paper: *J. https://doi.org/10.1021/acsnano.0c03139*, which demonstrates the singlet-to-triplet energy difference for multiple chiralities.

We thank the reviewer for introducing this informative paper. The $1/d^2$ dependence is also observed in singlet-triplet energy separation ΔE_{S-T} extracted from PL and optically detected magnetic resonance spectra¹. ΔE_{S-T} is 30 meV for tubes with a diameter of 1 nm which is smaller than ΔE_{11}^- and ΔE_{11}^{*-} for corresponding air-suspended tubes (Figs. 5c and 5d), presumably due to the difference in the dielectric environment. As suggested, we have added Supplementary Note 4 and the reference [44] in the revised manuscript.

7) The authors mention that they need to do the synthesis under UV illumination. Did they also test if the UV illumination itself could not cause such defects (thinking of the original UV - based doping work: DOI: 10.1126/science.1196382)

We appreciate this question from the reviewer. We have conducted experiments in the absence of iodobenzene and found that the UV illumination itself does not cause the formation of color centers (Supplementary Fig. 11). We therefore conclude that the color centers are generated by the chemical reaction between nanotubes and iodobenzene. The results from this control experiment along with a description are added as Supplementary Fig. 11 in the revised manuscript.

8) Several of the research works that the authors refer to regarding triplet excitons, show lifetimes in the microsecond regime, while here the authors see only a minor change in the lifetimes. Also for organic color centers synthesized for SWCNTs in dispersion, typically a much larger effect on the lifetimes is observed. Could the authors comment on this? Does this have to do with the large number of quenching sites present?

The reviewer is correct in that pure triplet excitons show long lifetimes in the microsecond regime [45, 52]. The shorter lifetime that we observe in the air-suspended tubes can be explained by hybridization of the singlet and triplet excitons, in which the lifetime can be determined by decay rates of the singlet states. Such hybridization has been observed by magneto-PL spectroscopy for functionalized (6,5) tubes [53].

Regarding the lifetimes, $\tau_1 = 69.1$ and $\tau_2 = 172.0$ ps for the air-suspended tube are comparable to $\tau_1 = 86$ and $\tau_2 = 171$ ps for solution-processed tubes [15]. This is reasonable because color centers are isolated and other defects should not affect the excitons trapped at the color centers. The comparable lifetimes also indicate that the vapor-phase reaction can form as high-quality color centers as the solution-processed methods.

The reviewer comments that a much larger effect on the lifetimes is observed, but the difference is the result of the longer E_{11} bright exciton lifetime for air-suspended tubes. The lower defect density results in the lifetime of E_{11} bright exciton $\tau_1 = 42.2$ ps for air-suspended tubes to be

longer than $\tau_1 = 26$ ps for solution-processed tubes. Since the color center lifetimes are comparable, the relative change in the lifetimes appears to be larger for the solution-processed tubes.

We have added these discussions in the second and third paragraphs on page 13 in the revised manuscript.

9) In figure 4c, could the authors also plot the spread on the data points (instead of or together with the error of the mean?). I think the spread is much more important. Furthermore, looking at figure 3, it would be interesting to provide a histogram plot of those distributions (perhaps in the SI) which would show that the distribution is quite asymmetric.

We agree that the spread is important. As suggested by the reviewer, we have plotted the spread on the data points and additionally analyzed the spread by converting scatters into histograms (Supplementary Figs. 4-6). Asymmetric distributions with long upper tails are observed in the subpeak ratio whereas lower tails are seen in the quenching degree for all the chiralities, indicating that relatively unreacted tubes are more abundant. We would like to keep Fig. 4c as is, since essentially the same information is contained in Fig. 3a and 3b. The analysis results of the distributions are shown in Supplementary Figs. 4-6 in the revised manuscript.

Response to Reviewer #2

The authors present a study on color-center and defect formation on suspended single wall carbon nanotubes through a gas phase doping process. Although the experiment is likely challenging and thus the large data set covering many chiral angles and diameters is impressive the extractable information seems limited by experimental choices. The most prominent achievements are the distinguishing of reactivity differences and mechanism winnowing. Some information may simply be missing due to the brevity of a communication template. I would support publication as a communication if the following items are addressed.

We thank the reviewer for the positive evaluation of our work. We are pleased that the reviewer found our efforts to be “impressive” and “prominent achievements”.

significant questions: 1. The primary comment is that this study appears to probe a relatively high level of functionalization rather than the sparse functionalization regime. The authors argue the opposite with the presented diffusion model, but I am concerned that the model a, does not consider the presence of some number of defects independent of functionalization or b, the potential for annihilation from the trion formation mechanism proposed later.

We acknowledge that the reviewer has concerns on our exciton diffusion model. Regarding the defects independent of functionalization, we would like to note that pristine air-suspended tubes can be considered defect free. Previous work has shown that PL intensity can be well

described by a defect-free model [28, 29], and therefore we believe that the presented diffusion model is a reasonable one.

Annihilation from trion formation should not affect our diffusion model. Such an effect is already included in the boundary conditions $n(z_d) = 0$.

These points have been clarified on line 161 of page 9 in the revised manuscript.

Is the distribution of I_0 intensities sufficiently monomodal to argue that the sparse regime is correct?

We do not quite understand the intent of this question since I_0 is the PL intensity for E_{11} peak before the reaction, but nevertheless we have provided scatter plots and histograms of I_0 below (Figs. R1 and R2). The distribution is found to be monomodal as the reviewer commented. In case the reviewer was referring to I_{11} which is the PL intensity for E_{11} peak after the reaction, we have also prepared scatter plots and histograms for I_{11} , also exhibiting monomodal distributions (Figs. R3 and R4).

Figure R1. Spectrally integrated PL intensity I_0 as a function of emission energy for experiments conducted with excitation energies of (a) 1.46 and (b) 1.59 eV and an excitation power of 10 μ W.

Figure R2. Histograms of spectrally integrated PL intensity I_0 for experiments conducted with excitation energies of (a) 1.46 and (b) 1.59 eV and an excitation power of 10 μ W. The bin width is chosen by Freedman Diaconis Estimator which takes into account data variability and size.

Figure R3. Spectrally integrated PL intensity I_{11} as a function of emission energy for experiments conducted with excitation energies of (a) 1.46 and (b) 1.59 eV and an excitation power of 10 μ W.

Figure R4. Histograms of spectrally integrated PL intensity I_{11} for experiments conducted with excitation energies of (a) 1.46 and (b) 1.59 eV and an excitation power of 10 μ W. The bin width is chosen by Freedman Diaconis Estimator which takes into account data variability and size.

2. Relatedly, the emission from induce color centers in an absolute sense should be parabolic-like, increasing from no functionalization to a maximum above which the increased number of defects leads to greater quenching. With the observed reactivity differences are all nanotube (n,m) structures observed in the same limit?

We agree with the reviewer that the emission intensity should show a maximum at a certain reaction time, but experimental data are consistent with all chiralities being in the increasing emission regime. We have observed the diameter-dependent subpeak ratios (Fig. 4c), which would not have been possible if the emission was near the maximum regime.

Can additional information be decoded from the distribution of observed spectra? Publication of the data set of pre and post functionalized observations could enable other researchers to explore such features if the authors argue that it is beyond the scope of this work.

We believe that we have analyzed the data set in considerable depth, but we agree with the reviewer that sharing our data could result in additional findings by other researchers. We plan to share the spectroscopic data in RIKEN data repository which is currently being built; in the meantime, we will be happy to share the data upon request. We have added the data availability statement on line 306 of page 15 in the revised manuscript.

3. Can the authors comment on how spatial inhomogeneity of functionalization may affect their observations and conclusions?

We thank the reviewer for bringing up this point. We recognize that the spatial inhomogeneity of functionalization may affect the estimation of the defect density. Our model for simulating an exciton density profile in a functionalized tube assumes that a defect is always formed at the center of the tube and other defects are created with even separations. This defect distribution results in the smallest PL intensity among all different distributions with the same number of defects, and thus the estimated defect density is a lower bound. We have added the discussion in Supplementary Note 1 of the revised manuscript.

Are functionalization sites close to the trench ends observable? Or do these contribute a region of color centers observed as defects?

Yes and no; The functionalized sites close to the trench ends are observable by E_{11}^- and E_{11}^{-*} emission if the sites are color centers (Figs. 1e and 1f), whereas they are not observable if the sites are quenching sites because end quenching has already diminished the E_{11} intensity [Supplementary Note 7 in Otsuka *et al.*, *Nat. Commun.* **12**, 3138 (2021).].

The authors note that different trench widths were utilized but do not discuss this parameter.

We appreciate this important comment. For shorter tubes, end quenching effects become large [28, 29]. We disregarded this type of quenching process in the strain-induced reactivity model (Eq. 1) and the exciton diffusion model (Eq. 2), which could lead to under-evaluation of trapping at defects. To eliminate the end quenching effect, we have reanalyzed only the longer tubes suspended across trenches with the largest width of 3.0 μm in the revised manuscript. Although we found that the quenching ratios were not under-evaluated, we decided to revise Fig. 4c and 4d as this analysis is more consistent with our diffusion model. We have accordingly revised a sentence on line 150 of page 8. A difference in obtained values of η for quenching degree between the previous and the revised manuscripts is less than 7%, which does not alter our conclusion.

4. Experimentally the authors trigger functionalization by a UV source, but other works induce functionalization by E22 excitation, are the spectra static and repeatable when a tube is observed several independent times?

The PL time trace measurement included in the Supplementary Information in the original manuscript shows that E_{22} excitation does not trigger functionalization (Supplementary Fig. 7). There are temporal fluctuations in the intensity for E_{11}^- and E_{11}^{-*} , but they are intermittent blinking that can be explained by trapping and detrapping of surface charge [Supplementary Ref. 1]² and do not indicate irreversible reaction. We have added this discussion to the caption of Supplementary Fig. 7 in the revised Supplementary Information.

5. Spectra in Figure 1C are taken at a power of 10 uW, but most data is reported for 100 uW excitation. The observation of trion and triplet phenomenon could be tied to the power of observation. Is the result independent of power? Was the experiment limited by practicality concerns to such conditions? Addition of short discussion is in order.

We thank the reviewer for pointing this out. As suggested, we have carried out power dependent PL spectroscopy measurements (Supplementary Fig. 9). The experiments for quenching ratio performed at a power of 10 μ W is in the linear regime, and therefore the results should be independent of power.

The dependence shows that E_{11} and E_{11}^- become sublinear above 20 μ W, but the data for subpeak ratio were taken at a power of 100 μ W. As the reviewer suspected, the signal-to-noise at the excitation power of 10 μ W is insufficient to accurately extract the peak intensity and position. We recognize the limitation and thus use the quenching ratio instead of the subpeak ratio to estimate the defect density. To address the reviewer's comment, we have added Supplementary Fig. 9 and an explanation in Supplementary Note 2 of the revised manuscript.

Errata: Figure S4 the E11- label is placed over a line feature that is very close to 1600 cm-1 from the E11 emission. The vastly narrower linewidth might also imply that it is a Gband feature.*

We thank the reviewer for noticing this error. We have corrected the positions of the labels for E_{11}^- and E_{11}^{-*} in Supplementary Fig. 7. A sharp line at 0.89 eV is likely E_{11} emission from a tube with another chirality. This line is not a G band feature because it should appear at 1.39 eV for an excitation energy of 1.59 eV. Accordingly, we have edited the caption of Supplementary Fig. 7 in the revised manuscript.

Many of the emission line scans in Figure S2 appear to show 3 to 4 features. Are these attributable to additional defect configurations? Raman features?

As the reviewer pointed out, in addition to E_{11} , E_{11}^- , and E_{11}^{-*} , some spectra show a fourth peak. It is reasonable that the fourth peak at an energy lower than E_{11}^{-*} is emission from color centers with a different binding configuration. Because the excitation energy is at 1.59 eV, and we do not expect any Raman features in this energy range. To address this comment, we have added a sentence to the caption of Supplementary Fig. 2 in the revised Supplementary Information.

Figure S5B: density is misspelled on the x-axis.

We thank the reviewer for the careful review of our manuscript including the Supplementary Information. As pointed out, we have corrected the misspelling in Supplementary Fig. 8 in the revised manuscript.

Response to Reviewer #3

In the manuscript, “Formation of Organic Color Centers in Air-Suspended Carbon Nanotubes Using Vapor-Phase Reaction,” the authors introduce a new functionalization technique that adds organic color centers to the single-walled carbon nanotube (SWCNT) sidewall. Covalent functionalization of air-suspended SWCNTs occurs via a photochemical reaction with adsorbing precursor molecules, which the authors say is desirable to the standard solution-phase chemistry that can contaminate the tube surface and alter photoluminescence (PL) characteristics due to interactions between the nanotube (NT) and local environment. Observations from roughly 2000 individual SWCNTs of varying chiralities revealed the formation of two defect-related emission peaks, referred to as E11- and E11-, that appear at lower energies from the E11. In conjunction with experimental data, the authors developed theoretical models and simulations aimed at uncovering the nature of the defect states. Time-resolved PL measurements were also acquired to investigate the dynamics of the defect states.*

We sincerely thank the referee for accurately summarizing the results.

The concepts discussed in the manuscript have been of significant interest to the NT community for some time. Indeed, in the last few years several studies have demonstrated 1) the unique single photon emitting properties of quantum defect-decorated SWCNTs, 2) ways to circumvent the creation of multiple photoactive defect states and 3) approaches to minimize perturbations to the integrity of the NT system due to interactions with the local environment. This manuscript contributes to the same overall body of research in this topical area, but besides providing an alternative functionalization method for creation of color-centers, it is not clear that it has the high novelty and breadth of impact expected of publications in Nature Communications. Indeed, the main conclusions that were drawn from theoretical modeling and simulations are not new or unique when compared to previous studies of defect-decorated SWCNTs: diameter-dependent defect emission shifts have been observed and calculations of defect density have been performed. The manuscript, although interesting, largely presents itself as simply another way to covalently form quantum defect sites on SWCNTs. Therefore, the scope seems to be focused on the narrow range of scientists who are interested in nanotube defect photophysics.

The significance of our work lies in the fact that color centers can now be introduced into air-suspended nanotubes. These pristine tubes can be considered defect free [28, 29], allowing for investigation of color centers in an ideal system. Not only would it lead to deeper understanding of color center properties, but it may also potentially lead to improving quantum emitter performance. In solution-processed tubes, existence of naturally formed defects complicates such investigations. We are able to obtain the defect density using a simple model, whereas estimation of defect density for solution-processed tubes is difficult. In the only work that mentions the defect density, the estimate spans two orders of magnitude [24].

We would also like to point out that other reviewers recognize the significance of our work. Reviewer #1 states that “*D. Kozawa et al present a new way of synthesizing organic color centers on air-suspended SWCNTs. While such color center synthesis is very well known for SWCNTs dispersed in solutions, a comparable synthesis strategy for air-suspended SWCNTs was not yet available.*” Reviewer #2 also mentions that “*the experiment is likely challenging and thus the large data set covering many chiral angles and diameters is impressive*”.

Nevertheless, to further clarify the significance of our work, we have revised paragraph 2 of in the Introduction section to include the following sentences.

“In comparison to solution-processed tubes that have naturally formed quenching sites [30], the air-suspended nanotubes can be considered defect free except for the tube ends [28, 29]. Such a system should provide an ideal platform for investigating exciton physics in color centers [14, 24], opening up new opportunities in nanoscale photonics using one- and zero-dimensional hybrid structures.”

In addition, the experimental data included in this manuscript is often presented without appropriate discussion or explanation, leaving the reader with many fundamental questions (see comments below). The manuscript would benefit from a paragraph or two that clearly describes why this work is important, what these results mean for the NT community, and how improvements can be made to realize the full potential of the vapor-phase reaction method.

We thank the reviewer for the criticisms and suggestions. We have addressed all comments in the detailed response below.

Comments: 1a) The E₁₁- and E₁₁- emission features appear significantly weaker than the E₁₁ emission feature (Figures 1 and 2). Previous studies of quantum defect-decorated SWCNTs have shown that defect emission features will dominate the SWCNT PL spectrum because excitons are rapidly funneled into the defect states. The authors do not address this discrepancy directly in the manuscript. Why do the vast majority of E₁₁- and E₁₁-* features appear so weak?*

We thank the reviewer for pointing this out. First, we would like to remind the reviewer that typical solution-processed tubes have smaller diameter than the air-suspended tubes. It is not possible to directly compare these tubes because the quantum yields of both E₁₁ and the additional peaks are diameter dependent. The shallower potential depth with the larger diameter for air-suspended SWCNTs results in the lower quantum yield of emission from color centers [15, 22], which reduces the subpeak ratio.

The lower quenching site density in air-suspended nanotubes also contributes to reduction of the subpeak ratio because E₁₁ emission is stronger. We emphasize that the initial quenching site density differs from solution-processed tubes. Pristine air-suspended tubes can be considered defect free [28, 29], whereas solution-processed tubes have naturally formed defects

[30] with a typical density of $8.3 \mu\text{m}^{-1}$. E_{11} emission for solution-processed tubes is weak because of the existing defects and thus their subpeak ratio is larger.

The reasons above make it difficult to quantitatively compare the subpeak ratio between these SWCNTs. We would also like to note that the study of vapor-phase reaction has just begun and there is room for improving the PL quantum yield as detailed in the response to Comment 1d. To address this point, we have added these explanations in Supplementary Note 3 in the revised manuscript.

1b) Are the relatively weak features of E_{11} - and E_{11} - a consistent shortcoming of the vapor-phase reaction method?*

We feel that it is too early to say that the relatively weak emission is a consistent shortcoming of the vapor-phase reaction method. This work presents the first demonstration of functionalization for air-suspended SWCNTs, and we believe that these quantum yields can be improved as detailed in the response to Comment 1d.

1c) What do the relatively weak features of E_{11} - and E_{11} - (compared to E_{11}) imply about the underlying photophysics of the exciton in the suspended NTs (versus the solution phase NTs)?*

Regarding the underlying photophysics of the excitons, please refer to our response to Comment 1a and Supplementary Note 3.

1d) Can the vapor-phase reaction method be competitive with established NT sidewall functionalization techniques if it consistently yields weak defect emission features? A detailed discussion of these topics is needed to improve the manuscript.

As answered in Comment 1b, we feel that it is too early to conclude that the vapor-phase reaction consistently yields weak defect emission features. We believe that the method can become competitive by improving reaction conditions.

We propose to optimize the photochemical reaction and precursor molecules. The formation of color center density can be adjusted by optimizing the duration of reaction time, ideally creating a single color-center per nanotube. As observed in the control experiments of the photochemical reaction in the absence of iodobenzene, UV irradiation itself introduces quenching sites (Supplementary Fig. 11). Suppression of this process could be possible by optimizing the power and the energy of UV light. By introducing deeper traps by different precursor molecules, quantum yield for E_{11}^- may be improved [22, 23].

We have added the discussion on line 253 of page 13 and Supplementary Fig. 11 in the revised manuscript.

2a) On page 5, the authors note that spatial overlap between E_{11} emission and defect emission is “expected.” The authors need to explain and clarify why they think so? Organic color centers should appear more localized than typical E_{11} emission (even after accounting for diffraction limited spots).

We agree with the reviewer that organic color centers should appear more localized than typical E_{11} emission. What we meant was that the spatial profile of the E_{11} emission indicates the location of the nanotube and the additional peaks are observed along this tube. To clarify this point, we have corrected the sentence on line 99 of page 6 in the revised manuscript.

2b) Is it generally true that there appears to be more PL localization for E_{11} - emission than for E_{11}^* emission (refer to Figures 1e and 1f)?

We thank the reviewer for raising this point. The E_{11}^- emission is not always more localized than E_{11}^{*-} as shown in the PL intensity maps (Fig. R5), and we frequently observe comparable localization for the two peaks.

Figure R5. A PL spectrum (a) intensity maps of (b) E_{11} , (c) E_{11}^- , and (d) E_{11}^{*-} emission from a (10,5) tube with an excitation energy of 1.59 eV and a power of 100 μ W where the intensity is spectrally integrated over each emission peak.

3) PL quenching seems to be an issue with the vapor-phase reaction method. Can the authors comment on the ability to control whether a defect created via the vapor-phase reaction method becomes an organic color center or quenching site?

We thank the reviewer for this interesting question. This point will be very useful for improving reaction conditions in future work. We should be able to use the difference in the reactivity for color centers and quenching sites (Figs. 4c and 4d), by choosing tubes with smaller diameters. According to the negative control experiments under UV illumination in the absence of iodobenzene, optimizing the illumination conditions should allow for preferential formation of color centers. We have added this discussion on line 157 of page 8 and Supplementary Fig. 11 in the revised manuscript.

3a) There is a brief discussion on page 8 that states, “...the formation of color centers is more responsive to strain compared to quenching sites.” This is a very interesting statement, but the authors do not expand on it. A more detailed discussion as to why some defects become color centers and other defects become quenching sites would seem to be required.

We thank the referee for bringing up this point. Quenching sites are simultaneously formed by the irradiation of UV light as we mentioned in the response to Reviewer#1’s comment 7, whereas color centers are introduced by the reaction with iodobenzene. We have added the discussion on Supplementary Fig. 11 in the revised Supplementary Information.

4) It is unclear how the scattered data points in Figure 3 were extracted from the PL images. Was this an average value? Peak value?

We thank the reviewer for raising this point. The spectra were taken at the center of nanotubes and spatial mapping was not conducted as mentioned in our response to Reviewer #1’s Comment 1b. We have accordingly added the descriptions to the Materials and Methods section in the revised manuscript. The values in Fig. 3 are neither average nor peak values but were extracted by spectrally integrating the intensity as already described on line 118 of page 7.

How are PL intensity variations across the NT accounted for?

We would like to bring up to your attention that this is already described in line 126 of page 7: “The large variation of the ratio is likely caused by multiple factors including inhomogeneity among SWCNTs, temporal fluctuations in intensity (Supplementary Fig. 7), and positions of the defects [34].”

5) Figures 5a and 5b appear very busy and the trends are difficult to follow, largely due to the chirality labels on the plot. Remaking the figure such that a color-coded key is included to identify the chiralities would be helpful.

We thank the reviewer for this advice. As the reviewer suggested, we have revised Fig. 5a and 5b.

6) Did the authors attempt to collect separate time decays for the E11- and E11- features? Insight into the potential similarities or differences in PL dynamics for each defect feature could be very meaningful.*

We recognize the importance of the experiments that the reviewer mentioned, however, we had difficulties in performing the measurements at the time because of insufficient detector sensitivity for color center emission. In the future, we plan to introduce a higher sensitivity detector, which would allow for collecting time decays of the E_{11}^- and E_{11}^{-*} emission.

7) *The single paragraph discussing the PL lifetimes of the E₁₁- and E₁₁-* defect states lacks depth. There is no discussion of PL dynamics beyond simply stating the fitting parameters and time constants. On page 11, the authors state: “The fast component is assigned to the decay of bright excitons whereas the slow component reflects the dynamics of dark excitons.” While this statement is true, what does this mean with regards to the organic color centers. How are bright and dark excitons interacting with these defect sites? It might be too early to provide an exact explanation, but a discussion on the potential dynamics would be useful.*

We thank the reviewer for raising this point. As suggested, we have added the following discussion on line 233 of page 12.

“The biexponential behavior for E₁₁ and E₁₁⁻ emission is consistent with previous reports for solution-processed samples [15, 22, 49], where the decay for E₁₁ can be understood by a three-level model including bright, dark, and ground states [31, 50]. Both bright and dark excitons are populated for excitation in resonance with E₂₂ [50, 51] and inter-state transition takes place between the bright and dark states. The biexponential decay for color center emission can be similarly understood by the three-level model, as another dark state associated with color centers lies below the bright state [23] and the E₁₁⁻ manifold can be considered as independent of the E₁₁ states [15].”

8) *The use of the label “organic color centers” often is suggestive of single-photon characteristics. Do the authors have any evidence that these defect-decorated air-suspended SWCNTs are in fact single photon sources at room temperature? Or is the use of “organic color centers” simply based on the similarity to established work?*

We acknowledge that the phrase “organic color centers” sometimes implies single-photon characteristics, but we use this phrase literally; dopant states that are introduced with an organic precursor. This notation is widely used in established work that does not present single-photon characteristics [10, 19].

REVIEWER COMMENTS

Reviewer #1 (Remarks to the Author):

I would like to thank the authors to carefully address all the comments by the reviewers and make appropriate changes, including new figures in the supplementary information that evidence previous unclear statements.

I think the authors did a very good job in improving their manuscript, and making it more appropriate for a high-ranked journal like Nature Communications.

With these changes, I would like to recommend this manuscript for publication in Nature Communications.

Reviewer #2 (Remarks to the Author):

The authors address most of the questions raised in review adequately, and if the following are addressed I think publication is supportable.

1. I am still not convinced of the defect free initial condition or the low defect regime claim.

Evidence against this claim is Figure R1 showing in absolute units the intensity spread of the emission from individual nanotubes. These initial intensities vary by a factor of $\sim 10X$ or more depending on the (n,m) . While experimental collection factors may account for this, although then those same factor would necessarily need to be repeatable for the calculated ratios, a plot of $(I_0 - I_{11})/I_0$ versus I_0 should be shown. In the limit of a defect free initial condition the random placement of defects should result in a scatter plot without a consistent trend. If the brighter/dimmer nanotubes are affected by defects then the degree of color center observation &/or quenching would be a function of the initial brightness.

2. I am surprised that the exciton density in Fig 1e, calculated for the 1 um^{-1} defect density at $\pm 0.5 \text{ um}$ from the center is so so low, ~ 0.5 the no defect case. The calculated diffusion length from the values given is $\sim 330 \text{ nm}$, which would be nice to just state. Looking at the literature this actually does appear consistent with Phys. Rev. B 85, 085434.

After the remarks above were submitted I was asked by the editor to specifically comment on the response to reviewer #3's review due to that reviewer's unavailability. Originally provided editorially I was later asked to also add these remarks to the review to be helpful to the authors. The text of this editorial evaluation is below:

Taking a look at the response, the authors responded to each of the reviewers comments in a reasonable and expected manner (i.e., by rewriting and assigning unknowns to future work rather) for those comments that were clarifications or related to emphasis of discussion (most questions). As a reviewer I personally would be satisfied although not by a lot by such responses.

The potential exception would be the response to 1A. This is the main technical question in my mind raised by the other reviewer, and is essentially along the lines of what I am asking in my review too; why are the unreacted intensities so variable and the reacted intensities so weak? It impacts whether both the "defect free" initial assertion and relative number of color centers produced versus quenching sites are within the regime necessary to be truly interesting and for the analysis given. The author's argument in response is potentially correct but weak in my opinion. To dump such an important point into supplementary note 3 is questionable, especially since the reviewer directly notes that it is avoided being discussed in the manuscript in the first place.

I would say whether the response to 1a is acceptable is, or will be, an editorial decision. The 1st reviewer will have had it highlighted as an issue for them to ask about in this second round, so may have in their rereview in which case it would be a strong signal it is insufficiently addressed to date. However, while it is possible, I consider it unlikely that the authors would consent to what is likely to be open ended new experimental work to address this complaint, as it would likely require repeating what is a boundary pushing and probably tedious experiment, so I would anticipate all responses to the comment will be through rewriting and an editorial decision will be required at the end of the next round of review as to whether enough evidence/argument has been offered.

Reviewer #3 (Remarks to the Author):

In this revised version of the manuscript the authors have done an adequate job of addressing the majority of the technical and scientific concerns. Perhaps this reviewer will have to just disagree with the other reviewers as to the impact of the work presented. Clearly the manuscript reports some interesting and novel results: a new method to create organic color centers on NTs in a more controlled environment.

However, at least so far, the shallow nature of the traps leading to simultaneous E11 and E11-/E11* emission is a complete non-starter for any quantum science applications. The authors do not address how to get around this fundamental problem. Thus, I don't see the broad interest or impact in the manuscript outside of the NT photo physics community focusing on color centers.

Specifically - the following comment was supposed to guide the authors in this regard, and was not addressed.

"The manuscript would benefit from a paragraph or two that clearly describes why this work is important, what these results mean for the NT community, and how improvements can be made to realize the full potential of the vapor-phase reaction method."

Response to reviewer report for manuscript NCOMMS-21-07736B by Kozawa *et al.*

Response to Reviewer #1

I would like to thank the authors to carefully address all the comments by the reviewers and make appropriate changes, including new figures in the supplementary information that evidence previous unclear statements.

I think the authors did a very good job in improving their manuscript, and making it more appropriate for a high-ranked journal like Nature Communications.

With these changes, I would like to recommend this manuscript for publication in Nature Communications.

We are glad to hear that the reviewer recommends publication of our manuscript.

Response to Reviewer #2

The authors address most of the questions raised in review adequately, and if the following are addressed I think publication is supportable.

We thank the reviewer for the positive evaluation of the last revision.

1. I am still not convinced of the defect free initial condition or the low defect regime claim.

Evidence against this claim is Figure R1 showing in absolute units the intensity spread of the emission from individual nanotubes. These initial intensities vary by a factor of $\sim 10X$ or more depending on the (n,m) . While experimental collection factors may account for this, although then those same factor would necessarily need to be repeatable for the calculated ratios, a plot of $(I_0 - I_{11})/I_0$ versus I_0 should be shown. In the limit of a defect free initial condition the random placement of defects should result in a scatter plot without a consistent trend. If the brighter/dimmer nanotubes are affected by defects then the degree of color center observation &/or quenching would be a function of the initial brightness.

We thank the reviewer for further clarifying a question in the previous review and providing the insightful suggestion.

First, we would like to explain the origin of the initial intensity spread. PL intensity from suspended nanotubes depends strongly on their suspended lengths, and variation of the suspended length causes the intensity spread.

The length dependent PL intensity can be understood by a one-dimensional exciton diffusion model for defect-free nanotubes [29]. Excitons are generated upon photoexcitation and diffuse until either radiative recombination or quenching takes place. For small L , most excitons diffuse to unsuspended regions before radiative recombination and result in quenching. As L gets larger, less excitons reach the ends and more excitons radiatively recombine, resulting in the increased PL intensity. The length dependent PL intensity for various chiralities is shown in Fig. R1. We note that all SWCNTs in Ref. [29] are fully suspended over trenches which is confirmed by PL imaging.

Figure R1. Length dependence of PL intensity with an excitation power of $0.01 \mu\text{W}$ for six different chiralities. Lines are fits, and a fit is not shown for (10,9) because a reliable fit has not been obtained. (Originally published in Ishii *et al.*, *Phys. Rev. B* **91**, 125427 (2015).)

Such length dependence of PL intensity combined with distribution of L results in the intensity variation. The suspended length can vary when nanotubes fall to the bottom of the trenches during the growth, as seen in PL images (Supplementary Fig. 14).

This explanation is supported by additional simulations of I_0 . We assume log-normal distribution of L (Supplementary Fig. 15a) and compute I_0 for each L with defect free initial condition (Supplementary Fig. 15b). The simulation reproduces the experiments that show an asymmetric peak with a longer tail towards higher I_0 .

To examine if existence of initial defects can cause the spread of I_0 as the reviewer asserts, we have also conducted simulations of I_0 with various initial defect densities (Supplementary Fig. 16). None of the simulations show characteristic features of an asymmetric peak with a longer tail towards higher I_0 , and thus the assumption of the initial defects is inconsistent with the experimental results (Supplementary Fig. 13).

We have added this discussion as Supplementary Note 4 in the revised manuscript.

Next, we plot $(I_0 - I_{11})/I_0$ vs. I_0 as requested by the reviewer to consider the defect density regime. Experimental results of $(I_0 - I_{11})/I_0$ as a function of I_0 show a peak at $(I_0 - I_{11})/I_0 \sim 0.75$ with a large spread (Supplementary Fig. 17). We compare with simulations performed for both low and high defect density regimes. The quenching degree for defect density ρ_{add} is computed using nanotubes with the log-normal distribution of L . A simulation result with low $\rho_{\text{add}} = 1.5 \mu\text{m}^{-1}$ reproduces the experiments well (Supplementary Fig. 18a), resulting in a scatter plot without a consistent trend just as the reviewer noted. In contrast, a simulation result with high ρ_{add} shows saturation of $(I_0 - I_{11})/I_0$ at 1.0 (Supplementary Fig. 18b) being inconsistent with the experiments.

We have added this discussion as Supplementary Note 5 in the revised manuscript.

2. I am surprised that the exciton density in Fig 1e, calculated for the $1 \mu\text{m}^{-1}$ defect density at $\pm 0.5 \mu\text{m}$ from the center is so so low, ~ 0.5 the no defect case. The calculated diffusion length from the values given is $\sim 330 \text{ nm}$, which would be nice to just state. Looking at the literature this actually does appear consistent with Phys. Rev. B 85, 085434.

As suggested, we have added the diffusion length in line 172 of page 10 in the revised manuscript.

After the remarks above were submitted I was asked by the editor to specifically comment on the response to reviewer #3's review due to that reviewer's unavailability. Originally provided editorially I was later asked to also add these remarks to the review to be helpful to the authors. The text of this editorial evaluation is below:

Taking a look at the response, the authors responded to each of the reviewers comments in a reasonable and expected manner (i.e., by rewriting and assigning unknowns to future work rather) for those comments that were clarifications or related to emphasis of discussion (most questions). As a reviewer I personally would be satisfied although not by a lot by such responses.

We are glad to hear that the responses were made in a reasonable and expected manner.

The potential exception would be the response to 1A. This is the main technical question in my mind raised by the other reviewer, and is essentially along the lines of what I am asking in my review too; why are the unreacted intensities so variable and the reacted intensities so weak? It impacts whether both the "defect free" initial assertion and relative number of color centers produced versus quenching sites are within the regime necessary to be truly interesting and for the analysis given. The author's argument in response is potentially correct but weak in my opinion. To dump such an important point into supplementary note 3 is questionable, especially since the reviewer directly notes that it is avoided being discussed in the manuscript in the first place.

We hope that our response to Comment 1 has sufficiently addressed the reviewer's concern about the defect-free initial assertion and defect density regime explaining the variable unreacted intensities. Regarding the description on the subpeak ratio, we have edited and moved the text into line 223 of page 12 in the revised manuscript.

I would say whether the response to 1a is acceptable is, or will be, an editorial decision. The 1st reviewer will have had it highlighted as an issue for them to ask about in this second round, so may have in their rereview in which case it would be a strong signal it is insufficiently addressed to date. However, while it is possible, I consider it unlikely that the authors would consent to what is likely to be open ended new experimental work to address this complaint, as it would likely require repeating what is a boundary pushing and probably tedious experiment, so I would anticipate all responses to the comment will be through rewriting and an editorial decision will be required at the end of the next round of review as to whether enough evidence/argument has been offered.

We thank the reviewer for thoroughly considering our manuscript. As the first reviewer accepted our response and recommended publication, we hope that the additional computational work we performed has sufficiently addressed the reviewer's concern.

Response to Reviewer #3

In this revised version of the manuscript the authors have done an adequate job of addressing the majority of the technical and scientific concerns. Perhaps this reviewer will have to just disagree with the other reviewers as to the impact of the work presented. Clearly the manuscript reports some interesting and novel results: a new method to create organic color centers on NTs in a more controlled environment.

We sincerely thank the reviewer for the positively evaluating the revision and concisely summarizing the impact of our work.

*However, at least so far, the shallow nature of the traps leading to simultaneous E11 and E11-
/E11* emission is a complete non-starter for any quantum science applications. The authors do not address how to get around this fundamental problem. Thus, I don't see the broad interest or impact in the manuscript outside of the NT photo physics community focusing on color centers.*

We acknowledge that this is indeed an important point and would like to bring up your attention that we did address this comment in the previous response. We added line 262 of page 14: "By introducing deeper traps by different precursor molecules, quantum yield for E_{11}^- may be improved [22, 33]."

In addition, the shallow potential studied here is related to the relatively larger diameter of air-suspended tubes compared to typical solution-processed tubes as described in line 223 of page 12. Quantum yield of color center emission should be comparable if we use air-suspended tubes with smaller diameters. To include this point, we have added the following sentence in line 266 of page 14: “Because the quantum yield of color center emission depends on the diameter [22], SWCNTs with smaller diameters are suited for creating deeper trapping potential.”

We have also added the following sentence to the same paragraph to further clarify how improvements can be made. “It is desirable to develop vapor phase chemistry with these molecules for air-suspended nanotubes as the quantum yield in solution-processed nanotubes is improved by ~50% with a trapping potential deeper by 16 meV [22].”

Specifically - the following comment was supposed to guide the authors in this regard, and was not addressed.

"The manuscript would benefit from a paragraph or two that clearly describes why this work is important, what these results mean for the NT community, and how improvements can be made to realize the full potential of the vapor-phase reaction method."

We acknowledge that our previous response was unclear. As this comment was in the introductory paragraph of the report, we intended to address the comment through the response to individual specific comments by the reviewer. Below we summarize what we should have explicitly noted in the previous response.

Regarding why this work is important, we added the following sentences in line 47 of page 3: “In comparison to solution-processed tubes that have naturally formed quenching sites [30], the air-suspended nanotubes can be considered defect free except for the tube ends [28, 29]. Such a system should provide an ideal platform for investigating exciton physics in color centers [14, 24], opening up new opportunities in nanoscale photonics using one- and zero-dimensional hybrid structures.”

Regarding what these results mean for the NT community, two sentences that address this comment were already included in line 45 of page 3 and in line 51 of page 3: “Further development of quantum emitters with improved performance is expected if color centers can be introduced to as-grown air-suspended SWCNTs known for their pristine nature [28, 29].” “Existing methods, however, require liquid-phase reaction where solvents and surfactants will inevitably be in contact with the nanotubes, making them incompatible with air-suspended tubes. To combine the excellent optical properties of the air-suspended SWCNTs with these organic color centers, an intelligent design of chemical reaction is needed.”

Regarding how improvements can be made, we added the following sentences in line 256 of page 13: “Finally, we discuss how improvements can be made to take full advantage of the vapor-phase reaction. The formation of color center density can be adjusted by optimizing the

duration of reaction time, ideally creating a single color-center per nanotube. As observed in the control experiments of the photochemical reaction in the absence of iodobenzene, UV irradiation itself introduces quenching sites (Supplementary Fig. 11). Suppression of this process could be possible by optimizing the power and the energy of UV light. By introducing deeper traps by different precursor molecules, quantum yield for E_{11}^- may be improved [22, 33].” In addition, we have added more discussion in this round about enhancing quantum yield of color center emission in line 264 of page 14.

REVIEWER COMMENTS

Reviewer #1 (Remarks to the Author):

Dear Authors,

After my initial letter that supported the publication of this manuscript, I was asked by the editor to have a look at the remaining discussion points regarding the fact if the SWCNTs are pristine to start from or if they already contain a large number of defects – and if the changes to the manuscript that were made in the mean time are sufficient to address this question. In my opinion they are sufficient to warrant publication, however, I do agree that it would be better to emphasize the newly added sections in the SI more in the main text such that readers will be drawn to those sections in the SI.

Below, I give some advice on how to add changes to the manuscript to make it acceptable for publication.

On page 7, line 119 you refer to supplementary note 4 but do not specify what is exactly in that supplementary note and why you refer to it. You could e.g. change it by emphasizing it more with a couple of additional sentences, e.g. something like: “ As can be observed, the intensity dispersion is quite large, which is also observed for the suspended SWCNTs without functionalisation and can be attributed to the varying suspension length of the CNTs, from approx. 0.5-3 μ m. Indeed, as observed previously (refer to the previous paper on the length dependence) and from simulations we performed in this work (see Supplementary Note 4) the intensity distribution for the unfunctionalized SWCNTs corresponds to a log-normal length distribution centered around $L=0.78\mu$ m. After functionalisation, it can be observed that the ratio of I11- and I11 becomes broader towards smaller diameters (see also Figure S4), indicative of the smaller tubes having more color centers.”

So I propose to elaborate more in the main text (with your own wording of course) on the different parts you added in the SI.

Similarly, on line 121 you can add something like: Due to this broadly varying emission intensity for non-functionalised SWCNTs, it is however more scientifically relevant to investigate the reduction of emission intensity due to the functionalisation. (□ so highlight this is the more physically relevant parameter to compare, than the absolute intensity as such you “delete” the length-dependence).

I also think that in this section (lines 123-125) it is important to explain better what you mean, as now I got confused on what was told: I e.g. do not see the trend with diameter in figure 3d? Perhaps you can also specify what is the effect of the resonance behavior on the intensities? And that that is the reason why you need to use the value of $(I_0-I_{11})/I_0$.

On lines 159-161 there you assume again that air-suspended tubes are defect free, while you could evidence it here with your supplementary note 4 – Supplementary Figure 16

(note that on page 13 in the SI, you wrongly refer to Supplementary Figure 17 instead of 16)

I think SI Figure 16 still requires a calculation without defect density or much lower defect density, as now for each of the calculations you have a number of tubes that don't show a PL intensity.

One should also be aware that experimentally, those tubes with 0 intensity are neglected, so that the curves cannot be compared directly with each other!! Please make a statement about that!

What is e.g. done with tubes that first emit, and afterwards don't emit anymore? Are they still in the statistics?

And please, refer to this again with a few more lines in the main text, and e.g. explain that from those calculations you know that the upper limit of defects on the tubes is ...

In case you would have the length of each of the tubes, it would be good to also show a plot of emission intensity (I_0) versus length of the CNTs instead of referring to older literature on 'other' samples.

Please also pay extra attention to the sections that were added now in the main text and the SI, as the English is much less polished than in the original version of the manuscript.

Reviewer #2 (Remarks to the Author):

As a reviewer, I am significantly frustrated by the great lengths to which the authors have obfuscated the wide variation in apparent suspended nanotube lengths. It still is non-obvious without deep reading of supplemental notes that the suspended length is not 3 microns for most data, and the figure that led to the question that brought this up was only provided in review to reviewers only and is not even available to me now as I deleted the file after the prior review.

The authors should clearly provide a scatterplot figure of the apparent length versus the trench width in the main paper. The assumption of a log normal distribution in my view is clearly inappropriate. Why would data from several discrete trench widths with different likelihood of suspension and suspended lengths be a semicontinuous log normal! Arguments based on generated simulations against a continuous function, when the real data must be dis-continuous by the combination of data from multiple trench widths is severely weakened.

The authors must also clearly state that most nanotubes are not fully suspended in the main text!

This does not necessarily imply that the results generally are incorrect, but the need to extract such vital information to evaluate whether experimental methods are consistent with the authors claims is not appreciated.

Response to Reviewer #1

1) After my initial letter that supported the publication of this manuscript, I was asked by the editor to have a look at the remaining discussion points regarding the fact if the SWCNTs are pristine to start from or if they already contain a large number of defects – and if the changes to the manuscript that were made in the mean time are sufficient to address this question. In my opinion they are sufficient to warrant publication, however, I do agree that it would be better to emphasize the newly added sections in the SI more in the main text such that readers will be drawn to those sections in the SI.

Below, I give some advice on how to add changes to the manuscript to make it acceptable for publication.

We thank the reviewer for the positive evaluation of the last revision with the “*sufficient to warrant publication*” remark. We have revised our manuscript following the reviewers’ suggestions, and we believe the manuscript has considerably improved.

2) On page 7, line 119 you refer to supplementary note 4 but do not specify what is exactly in that supplementary note and why you refer to it. You could e.g. change it by emphasizing it more with a couple of additional sentences, e.g. something like: “As can be observed, the intensity dispersion is quite large, which is also observed for the suspended SWCNTs without functionalisation and can be attributed to the varying suspension length of the CNTs, from approx. 0.5-3 μ m. Indeed, as observed previously (refer to the previous paper on the length dependence) and from simulations we performed in this work (see Supplementary Note 4) the intensity distribution for the unfunctionalized SWCNTs corresponds to a log-normal length distribution centered around $L=0.78\mu$ m. After functionalisation, it can be observed that the ratio of I₁₁- and I₁₀ becomes broader towards smaller diameters (see also Figure S4), indicative of the smaller tubes having more color centers.”

So I propose to elaborate more in the main text (with your own wording of course) on the different parts you added in the SI.

We thank the reviewer for this constructive suggestion. We have accordingly added a description for Supplementary Note 1 (Supplementary Note 4 in the previous version of the manuscript) on page 6, line 113 in the revised manuscript: “Large dispersion is observed (Supplementary Figs. 12, 13, and 14), which can be attributed to various suspended lengths (Supplementary Note 1). Many SWCNTs are not fully suspended as observed in PL images (Supplementary Fig. 15), and their suspended lengths are likely shorter than the trench widths. Indeed, the intensity dispersion is well reproduced by simulations of length dependent PL intensity [28, 29] assuming a log-normal length distribution [34] centered at 0.78 μ m (Supplementary Fig. 16), indicating that most nanotubes have lengths ranging from 0.5 to 1.0 μ m.” We have revised the next few paragraphs on pages 6-7 for readability.

In addition, we have added the following descriptions referring to the other parts added in Supplementary Information:

On page 9, line 187, regarding Supplementary Note 2: “although the higher excitation power used to obtain the subpeak ratio could lead to underestimation of the reactivity for the color centers (Supplementary Fig. 18 and Note 2)”

On page 11, line 228, regarding Supplementary Note 3 (Supplementary Note 1 in the previous version of the manuscript): “The estimated defect density represents a lower bound because of the assumption in the simulations that defects are uniformly distributed in nanotubes (Supplementary Note 3).”

On page 11, line 230, regarding Supplementary Note 4 (Supplementary Note 5 in the previous version of the manuscript): “We also compare the distribution of $(I_0 - I_{11})/I_0$ between experiments and simulations (Supplementary Note 4). The experiments are well reproduced by a simulation assuming the log-normal length distribution and $\rho = 1.5 \mu\text{m}^{-1}$ (Supplementary Figs. 19 and 20a). In comparison, a simulation with a high defect density of $\rho = 10 \mu\text{m}^{-1}$ cannot reproduce the experimental results (Supplementary Fig. 20b).”

3) Similarly, on line 121 you can add something like: Due to this broadly varying emission intensity for non-functionalised SWCNTs, it is however more scientifically relevant to investigate the reduction of emission intensity due to the functionalisation. (so highlight this is the more physically relevant parameter to compare, than the absolute intensity as such you “delete” the length-dependence).

We thank the reviewer for this suggestion. We have added sentences that highlight why we use the quenching degree rather than the absolute intensity on page 7, line 150 in the revised manuscript: “In a manner similar to the subpeak ratio, we take the quenching degree $(I_0 - I_{11})/I_0$ as a more physically relevant quantity for comparison between different nanotubes with various suspended lengths and chiralities.”

4-1) I also think that in this section (lines 123-125) it is important to explain better what you mean, as now I got confused on what was told: I e.g. do not see the trend with diameter in figure 3d?

We thank the reviewer for bringing up this point. The trend on diameter dependence is less clear in Fig. 3d due to overlap of dots in the scatter plot, but the trend becomes more apparent by using average values plotted against the nanotube diameter, as shown in Fig. 4d.

4-2) Perhaps you can also specify what is the effect of the resonance behavior on the intensities? And that that is the reason why you need to use the . value of $(I_0 - I_{11})/I_0$.

We agree with the reviewer that the resonance to the excitation energies affects PL intensity. We have accordingly added sentences on page 7, line 141: “It should be noted that the chirality

dependent E_{22} energy (Supplementary Fig. 5) results in emission intensity differences, since we fix the excitation energy either at 1.46 or 1.59 eV. Taking the ratio cancels out the chirality dependent E_{11} emission intensity, allowing for direct comparison between different chiralities. The intensity dispersion can also be caused by resonance shifts due to initial strain generated during growth and inhomogeneity of dielectric environment, but PL intensity variations should be insignificant since energy shifts are typically within ± 10 meV [29]. Although small, the effects of these variations are likewise reduced by taking the ratio.”

5) On lines 159-161 there you assume again that air-suspended tubes are defect free, while you could evidence it here with your supplementary note 4 – Supplementary Figure 16 (note that on page 13 in the SI, you wrongly refer to Supplementary Figure 17 instead of 16) I think SI Figure 16 still requires a calculation without defect density or much lower defect density, as now for each of the calculations you have a number of tubes that don't show a PL intensity.

We appreciate this suggestion from the reviewer. We have corrected the figure reference as the reviewer pointed out. As requested by the reviewer, we have also conducted a simulation with $\rho_{\text{mit}} = 0 \mu\text{m}^{-1}$ (Supplementary Fig. 17) and added descriptions on page 9, line 196 in the revised manuscript: “We use the fact that pristine air-suspended nanotubes are defect free except for end quenching [28, 29]. Although the nanotubes used in this study have varying suspended lengths (Supplementary Note 1), the experimentally obtained distribution of I_0 (Supplementary Fig. 12) is consistent with a simulation assuming defect free nanotubes (Supplementary Fig. 16). In comparison, the experimental results cannot be reproduced if initial defects are included in the simulations (Supplementary Fig. 17)”.

6-1) One should also be aware that experimentally, those tubes with 0 intensity are neglected, so that the curves cannot be compared directly with each other!! Please make a statement about that!

We thank the reviewer for pointing this out. Pristine nanotubes with a peak height of less than 500 counts/s are excluded in the statistics and not used for further measurements. We have accordingly added a sentence on page 17, line 372: “Pristine nanotubes with an E_{11} peak height of less than 500 counts/s are excluded in the statistics and not used for further measurements.” We have also added a statement on page 6, line 120 in the revised manuscript: “We note that the simulations cannot be directly compared to experiments for nanotubes with no detectable PL, but the fraction of tubes is negligible at low intensities for the distribution reproducing the experimental data.”

6-2) What is e.g. done with tubes that first emit, and afterwards don't emit anymore? Are they still in the statistics?

To obtain the quenching degree, we include functionalized nanotubes that first emit and afterwards do not emit in the statistics (Figs. 3c, 3d, 4d, and 4f). To obtain the subpeak ratio (Figs. 3c, 3d, and 4c) and peak positions (Fig. 5a and 5b), PL spectra with sufficient signal to noise are needed. We therefore only use PL spectra with an E_{11} peak height of more than

400 counts/s for functionalized nanotubes. To clarify this point, we have added sentences on page 17, line 374: “We also exclude functionalized nanotubes with an E₁₁ peak height of less than 400 counts/s.”

7) And please, refer to this again with a few more lines in the main text, and e.g. explain that from those calculations you know that the upper limit of defects on the tubes is ...

We thank the reviewer for this advice. We have accordingly added a sentence on page 9, line 200 in the revised manuscript: “In comparison, the experimental results cannot be reproduced if initial defects are included in the simulations (Supplementary Fig. 17), and we therefore estimate the defect density in pristine SWCNTs to be much less than 0.25 μm⁻¹.”

8) In case you would have the length of each of the tubes, it would be good to also show a plot of emission intensity (I₀) versus length of the CNTs instead of referring to older literature on 'other' samples.

We acknowledge that the length dependence of PL intensity could enable validation of the defect-free initial condition as was done in the older literature [28, 29]. Unfortunately, we do not have the data because it is unrealistic in terms of the time required to conduct PL imaging of more than 2000 individual nanotubes on a Si substrate. We estimate that it takes ~25 min to obtain a PL image with 31 by 31 pixels, which would correspond to ~37 days for obtaining 2000 PL images on a Si substrate and ~293 days for all the 8 substrates in total. This does not include any sample exchange time, any maintenance required, nor any analysis time.

9) Please also pay extra attention to the sections that were added now in the main text and the SI, as the English is much less polished than in the original version of the manuscript.

We thank the reviewer for this advice. We have carefully edited the sentences that were added in the series of review rounds. In particular, we have revised the text on pages 14-16, lines 286-330 to improve readability.

Response to Reviewer #2

1) As a reviewer, I am significantly frustrated by the great lengths to which the authors have obfuscated the wide variation in apparent suspended nanotube lengths. It still is non-obvious without deep reading of supplemental notes that the suspended length is not 3 microns for most data, and the figure that led to the question that brought this up was only provided in review to reviewers only and is not even available to me now as I deleted the file after the prior review.

We acknowledge the reviewer’s concern. To clarify that the suspended length is not 3.0 μm for most data, we have added the following sentences to the revised manuscript: On page 6, line 115, “Many SWCNTs are not fully suspended as observed in PL images (Supplementary Fig. 15), and their suspended lengths are likely shorter than the trench widths. Indeed, the intensity dispersion is well reproduced by simulations of length dependent PL intensity [28, 29]

assuming a log-normal length distribution [34] centered at 0.78 μm (Supplementary Fig. 16), indicating that most nanotubes have lengths ranging from 0.5 to 1.0 μm .”

Regarding the figures shown in the prior responses, we have added scatter plots of I_0 as a function of emission energy in Supplementary Fig. 13 and histograms of I_0 for each chirality in Supplementary Fig. 14. We have also added a reference to the figures on page 6, line 113: “Large dispersion is observed (Supplementary Figs. 12, 13, and 14),”

2) The authors should clearly provide a scatterplot figure of the apparent length versus the trench width in the main paper. The assumption of a log normal distribution in my view is clearly inappropriate. Why would data from several discrete trench widths with different likelihood of suspension and suspended lengths be a semicontinuous log normal! Arguments based on generated simulations against a continuous function, when the real data must be discontinuous by the combination of data from multiple trench widths is severely weakened.

We agree with the reviewer that a log-normal distribution would be inappropriate to describe data from several discrete trench widths, but we use the data with a single trench width of 3.0 μm to determine the reactivity (Figs. 4c and 4d) and to determine the defect density (Fig. 4f). This point was already described on page 9, line 180: “We only include the tubes on the widest trenches with 3.0 μm width to reduce end quenching effects.” and on page 11, line 221: “We use the experimental data of tubes on the widest trenches with 3.0 μm widths”. Since the data with the other trench widths are excluded in these analyses, the comparison between the experiments and the simulations should be appropriate.

Regarding a scatter plot figure of the apparent length *versus* the trench width, we acknowledge that determining the apparent length of each nanotube would be useful. We do not have these data because it is unrealistic in terms of time required for such an experiment. We estimate the time to be ~ 37 days/substrate for PL imaging for 2000 individual nanotubes and ~ 293 days for all the 8 substrates in total.

3) The authors must also clearly state that most nanotubes are not fully suspended in the main text!

We appreciate this advice from the reviewer. We have added the following sentence on page 6, line 115 in the revised manuscript: “Many SWCNTs are not fully suspended as observed in PL images (Supplementary Fig. 15)”.

4) This does not necessarily imply that the results generally are incorrect, but the need to extract such vital information to evaluate whether experimental methods are consistent with the authors claims is not appreciated.

We believe that our manuscript has considerably improved by following the reviewers’ advice. In response to Comments 1 and 3, we have clarified that many nanotubes are not fully suspended. In response to Comment 2, we explain that only the data with a single trench width

is used for the analysis, which is consistent with the use of log-normal distribution in the simulations.

REVIEWERS' COMMENTS

Reviewer #1 (Remarks to the Author):

I do not have any further comments for this manuscript.

In my opinion, you have done all you could to provide the requested additional information and change the text accordingly. It is of course a pity that length/intensity information is not fully available, but I understand the complexity of the experiments and the time that would require.

So I support the publication of the manuscript.

Reviewer #2 (Remarks to the Author):

The responses to the two reviews are acceptable.